# Emergence and function of cortical offset responses in sound termination detection

**Magdalena Solyga, Tania Rinaldi Barkat***

Department of Biomedicine, Basel University, Basel, Switzerland

**Abstract** Offset responses in auditory processing appear after a sound terminates. They arise in neuronal circuits within the peripheral auditory system, but their role in the central auditory system remains unknown. Here, we ask what the behavioral relevance of cortical offset responses is and what circuit mechanisms drive them. At the perceptual level, our results reveal that experimentally minimizing auditory cortical offset responses decreases the mouse performance to detect sound termination, assigning a behavioral role to offset responses. By combining in vivo electrophysiology in the auditory cortex and thalamus of awake mice, we also demonstrate that cortical offset responses are not only inherited from the periphery but also amplified and generated de novo. Finally, we show that offset responses code more than silence, including relevant changes in sound trajectories. Together, our results reveal the importance of cortical offset responses in encoding sound termination and detecting changes within temporally discontinuous sounds crucial for speech and vocalization.

## Editor's evaluation

The work demonstrates specific neurophysiological cortical mechanisms for offset responses that are interesting in themselves. Two referees highlighted issues with the behavioural experiments that have been addressed in the revision. Reviewer #2 makes another minor suggestion that he authors might consider before publication of the final version.

*For correspondence:
tania.barkat@unibas.ch

Competing interest: The authors declare that no competing interests exist.

## Introduction

Offset-responsive neurons are present through the whole auditory pathway starting from the cochlear nucleus (CN; *Ding et al., 1999*; *Suga, 1964*; *Young and Brownell, 1976*) to the superior paraolivary nucleus (SPN; *Dehmel et al., 2002*; *Kulesza et al., 2003*), the inferior colliculus (IC; *Akimov et al., 2017*; *Kasai et al., 2012*), the medial geniculate body (MGB; *Anderson and Linden, 2016*; *He, 2001*; *He, 2002*; *Yu et al., 2004*), and the auditory cortex (ACx; *Qin et al., 2007*; *Recanzone, 2000*; *Scholl et al., 2010*; *Takahashi et al., 2004*). Multiple mechanisms have been proposed to produce offset responses (*Bondanelli et al., 2019*; *Bondanelli et al., 2021*; *Kopp-Scheinpflug et al., 2018*; *Xu et al., 2014*). Generally, it is thought that signals from the cochlea can generate offset responses in both the CN (*Suga, 1964*) and the SPN (*Kopp-Scheinpflug et al., 2011*), but strong offset responses have mainly been described in the SPN. This structure is considered specialized for offset response generation based on the strong inhibitory signal it receives during the sound and on the precise firing when the sound ends. The SPN sends then strong inhibitory inputs to the IC (*Kulesza and Berrebi, 2000*; *Saldaña et al., 2009*), which might further convey the signal to MGB. Offset responses in MGB and the ACx are generally thought to be driven by excitatory/inhibitory inputs from IC rather than by other neural mechanisms (*Kopp-Scheinpflug et al., 2018*). Recently, Bondanelli et al. argued for a role of recurrent A1 connectivity in shaping offset responses in cortex, and suggested that cortical offset responses could be generated at a higher level of recurrency (*Bondanelli et al., 2021*). However, de

novo generation or amplification of offset responses in MGB and ACx have not been demonstrated experimentally yet (*Bondanelli et al., 2019*; *He, 2003*; *Kasai et al., 2012*; *Yu et al., 2004*).

The perceptual significance of offset responses has been difficult to assess. They have been postulated to play a role in sound duration selectivity (*He, 2002*; *Qin et al., 2009*), in gap detection (*Syka et al., 2002*; *Threlkeld et al., 2008*; *Weible et al., 2014a*; *Weible et al., 2014b*), or in perceiving communication calls (*Eggermont, 2015*; *Felix et al., 2018*; *Kopp-Scheinpflug et al., 2018*). For example, Qin et al. showed that onset-only neurons in the primary ACx of cats could not discriminate duration and suggested that sustained and offset responses underlie discrimination of sound duration (*Qin et al., 2009*). In another study, Weible et al. demonstrated that the cortical postgap neural activity in mice is related to detecting brief gaps in noise (*Weible et al., 2014b*). Recently, it was shown that A1 transient offset responses contribute critically to encoding and perceiving sound duration (*Li et al., 2021*). Whether the increased neuronal activity of sound offset responses accounts for other perceptual skills is unclear.

Compared to onset responses, offset responses in the central auditory pathway are typically less prevalent (*Phillips et al., 2002*; *Pollak and Bodenhamer, 1981*; *Sołyga and Barkat, 2019*). They are generally weaker and slower than onset responses (*Qin et al., 2007*). At the cortical level, offset responses have been shown to cluster within the anterior auditory field (AAF) – a primary region of the ACx – where they have been observed in twice as many cells as in the primary auditory cortex (A1) (*Sołyga and Barkat, 2019*). Offset responses are also highly influenced by different sound parameters. For example, the amplitude, duration, frequency, fall-time, and spectral complexity of the sound have all been reported to influence auditory offset responses (*He, 2002*; *Scholl et al., 2010*; *Sołyga and Barkat, 2019*; *Takahashi et al., 2004*). However, no study has yet systematically addressed these influences. The involvement of the different nuclei of the central auditory system, as well as their cellular and circuit mechanisms, are thus poorly understood.

We combined behavioral experiments with optogenetics, electrophysiological recordings, and antidromic stimulation to better understand the role of cortical offset responses in sound perception and the properties that distinguish them from the subcortical ones. Our results reveal that AAF is highly specialized for processing information on sound termination and that minimizing its offset responses decreases the mouse performance to detect when a sound ends. By studying the influence of different sound parameters on AAF and MGB offset responses, we demonstrate that cortical offsets are inherited, amplified, and sometimes even generated de novo. First, we found that AAF, unlike MGB, shows a significant increase in offset response amplitudes with sound duration. Then, we report that white noise (WN) bursts – unlike pure tones – only evoke offset responses in AAF but not in MGB. Finally, we show that offset responses are present in AAF whenever a frequency component is removed from multifrequency stimuli and therefore may have a further role than solely coding silence.

Taken together, our findings suggest a particular involvement of AAF offset responses in sound termination processing and point to the importance of this cortical subfield for advanced processing such as tracking sound duration or detecting changes in frequency and level within temporally discontinuous sounds.

## Results
### Cortical offset responses improve sound termination detection

The perceptual significance of cortical offset responses has been difficult to assess. Indeed, confounds about perceiving a sound and its termination are intricately linked. Changing the neuronal activity of sound offset responses to evaluate its contribution in sound termination detection has not been tested. To assess the behavioral relevance of auditory offset responses, we developed a sound termination detection task in which mice expressing channelrhodopsin-2 (ChR2) in parvalbumin-positive (PV+) cells learned to detect the end of 9 kHz pure tones (PT; *Figure 1a*). As, at the cortical level, transient offset responses cluster within AAF rather than A1 (*Sołyga and Barkat, 2019*) and are absent in the averaged population activity of A1 neurons in response to 9 kHz PT in awake preparations (*Figure 1—figure supplement 1*), we decided to focus on AAF to assess the behavioral relevance of cortical offset responses. During the training, animals were placed in a cardboard tube with a speaker 10 cm away from their left ear. A piezo sensor attached to a licking spout was used to measure sound termination detection during the task. To accelerate the learning process, mild air puffs were given

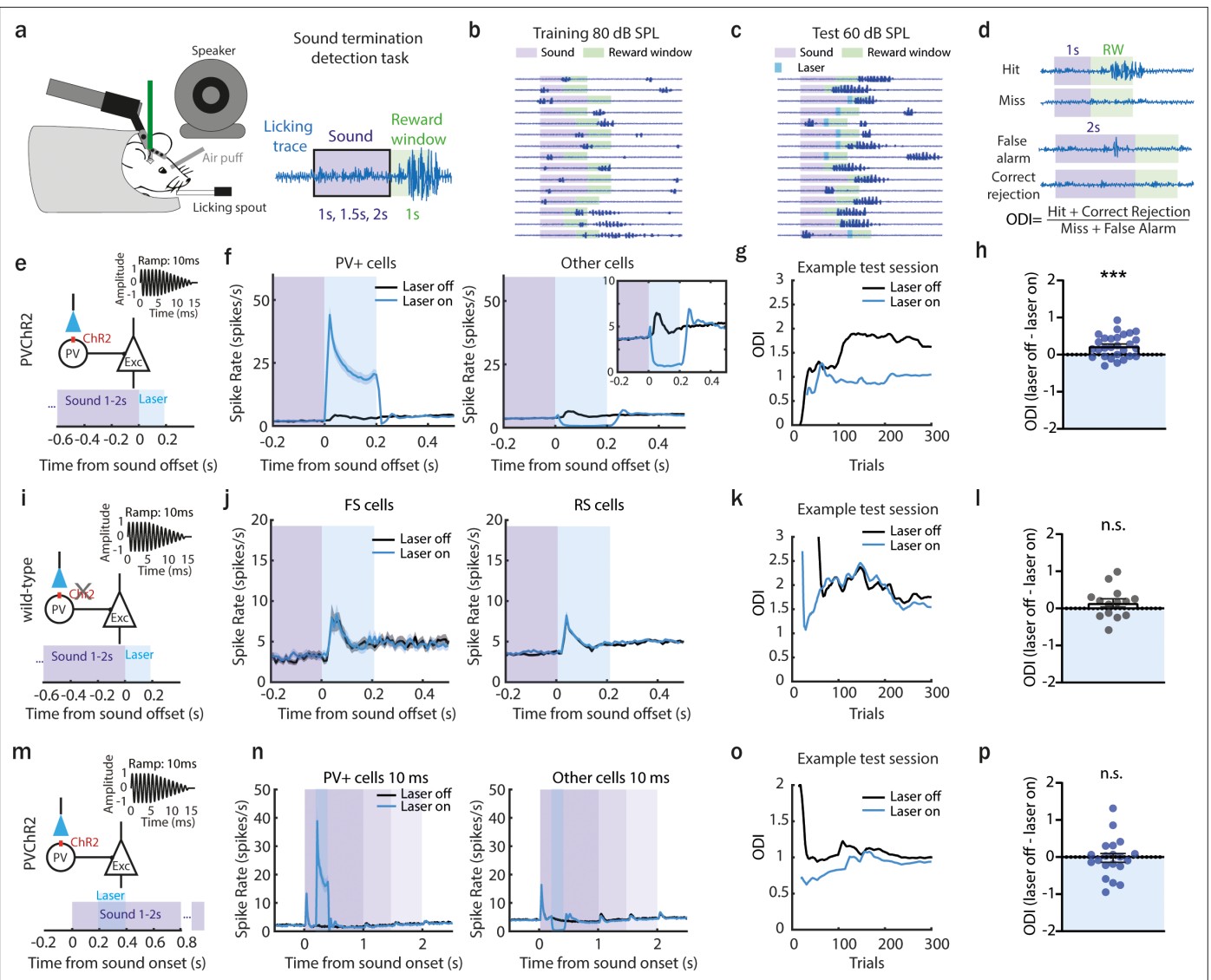

**Figure 1.** Preventing anterior auditory field (AAF) offset responses decreases the ability of a mouse to detect sound termination. (**a**) Illustration of the head-fixed behavioral setup. A piezo sensor attached to the licking spout measured behavioral response to the detection of sound termination (left). Schematic of the behavioral paradigm of offset detection task. The animal had to detect the termination of 9 kHz pure tone (PT) played with three different durations: 1, 1.5, and 2 s within a reward window of 1 s. (**b**) Licking traces of 15 consecutive trials for 1 example training session. Violet bar indicates sound, green – reward window. During training sessions, sounds were presented at 80 dB SPL. (**c**) Licking traces of 15 consecutive trials for 1 example test session. Violet bar indicates sound, green – reward window, blue – laser window. During test sessions, sounds were presented at 60 dB SPL. Additionally, in half of the trials, a laser light was applied for 200 ms following sound termination. (**d**) Definition of calculated offset detection index (ODI). (**e**) Schematic of laser manipulation. The laser light was used for 200 ms following sound termination in animals expressing channelrhodopsin-2 (ChR2) in parvalbumin-positive (PV+) cells. (**f**) Population activity (mean ± standard error of mean [SEM]) of PV+ ($n$ = 336 cells) or all other cells ($n$ = 2249 cells) following sound termination in laser-on (blue) and laser-off (black) trials, 28 sessions, 6 animals. Inset: zoom in. (**g**) ODI for an example session during the offset detection task. Blue and black lines represent the ODI for laser-on and laser-off trials, respectively. (**h**) The overall difference in ODI for laser-off and laser-on trials at the end of the training (mean ± SEM, ***p = 0.0005, $n$ = 28 sessions, 6 animals, 1 sample Wilcoxon test). (**i**) Schematic of control experiment: a PT of 9 kHz was used, and the laser was applied for 200 ms following sound termination in wild-type animals. (**j**) Population activity (mean ± SEM) following sound termination in laser-on (blue) and laser-off (black) trials in putative fast (left, $n$ = 103 cells, 14 sessions, 3 animals) and regular spiking neurons (right, $n$ = 930 cells, 14 sessions, 3 animals). (**k**) ODI for an example session during offset detection task. Blue and black lines represent the ODI for laser-on and laser-off trials, respectively. (**l**) Difference in ODI for control animals in laser-off and laser-on trials at the end of the training (mean ± SEM), p = 0.24, $n$ = 14 sessions, 3 animals, 1 sample Wilcoxon test. (**m**) Schematic of the experiment with laser suppression during the sound presentation: a 9 kHz PT was played, and the laser was applied for 200 ms following sound onset in animals expressing ChR2 in PV+ cells. (**n**) Population activity (mean ± SEM) in laser-on (blue) and laser-off (black) trials of PV+ cells (left, $n$ = 269 cells, 20 sessions, 4 animals) and other cells (right, $n$ = 1414 cells, 20 sessions, 4 animals). (**o**) ODI for an example session during offset detection task. Blue and black lines represent the ODI for laser-on

*Figure 1 continued on next page*

*Figure 1 continued*

and laser-off trials, respectively. (**p**) Difference in ODI for laser-off and laser-on trials at the end of the training (mean ± SEM), p = 0.67, *n* = 20 sessions, 4 animals, 1 sample Wilcoxon test.

The online version of this article includes the following figure supplement(s) for figure 1:

**Figure supplement 1.** Averaged poststimulus time histogram (PSTH; mean ± standard error of mean [SEM]) of anterior auditory field (AAF) (**a**) and A1 (**b**) neuron responses to 0.5 s long pure tone (PT; 9 kHz) played at 60 dB SPL.

**Figure supplement 2.** A detailed description of the sound termination detection task in parvalbumin (PV)-channelrhodopsin-2 (ChR2) mice.

**Figure supplement 3.** Evaluation of animal's behavior in sound termination detection task based on hit rate.

**Figure supplement 4.** Comparison of reaction times in sound termination detection task for trials with fast (0.01 ms) and slow (10.0 ms) fall-ramp and trials without and with laser stimulation.

when mice licked to sound onset. The duration of the tones was varied randomly (1, 1.5, or 2 s) to avoid a putative expected behavioral response at a fixed delay after sound onset. Mice were initially trained with a tone played at 80 dB SPL (sound pressure level) (*Figure 1b*, *Figure 1—figure supplement 2a*). A reaction time window was set to 3 s during initial training sessions and progressively shortened to 1 s in the final training sessions. Sound durations during initial training (1–4 days) were shorter (0.5, 1, and 1.5 s) and then switched to final durations (1, 1.5, and 2 s). The chance level of licks was calculated in a late window following the end of the reward window (*Figure 1—figure supplement 2b, d, h*). An increase in correct offset detection (hit rate), a decrease in reaction time over the training sessions (*Figure 1—figure supplement 2b*), and a significant difference between hit rate and licks at the chance level (*Figure 1—figure supplement 2e–g*) reflected successful learning after which animals were moved to the test phase (*Figure 1c*). Both offset detection index (ODI) (*Figure 1d*, see methods) and hit rate (*Figure 1—figure supplement 3*) were used to evaluate behavioral performance in the sound termination detection task.

After this training phase, animals underwent a craniotomy. On the following days, they were tested with tones played at 60 dB SPL. All our behavioral experiments with optogenetic manipulations were coupled with electrophysiological recording. For all behavioral sessions, we first identified AAF based on the tonotopic gradient obtained with electrophysiological recordings. We then placed an optical fiber above AAF (above the dura) and titrated the laser (473 nm) power to remove offset responses. Finally, we performed the behavioral session with these specific optogenetic parameters (optic fiber placement and laser power). We confirmed at the end of each session that offset responses were removed during the laser-on trials. No optical shield was used. In half of the trials (pseudorandomized each day), a laser light (473 nm) was delivered above AAF for 200 ms starting at sound termination (*Figure 1e*) to activate PV+ cells (*Figure 1f*) and to significantly reduce offset responses in non-PV+ cells (*Figure 1f*). Comparing the animals' performance during laser-on and laser-off trials showed that preventing AAF offset responses significantly decreased the performance to detect sound termination (*Figure 1g, h*, Δ ODI (off–on) = 0.23 ± 0.06). Interestingly, reaction times were significantly longer in laser-on compared to laser-off trials (*Figure 1—figure supplement 4*). To control that the light itself, without ChR2, was not causing any change in behavioral performance, we repeated the same experiments in wild-type animals (*Figure 1i*). The laser alone had no effect, neither at the neural (*Figure 1j*) nor at the behavior level (*Figure 1k,l*, Δ ODI (off–on) = 0.15 ± 0.11). Finally, we have performed experiments in PV-ChR2 animals where the laser light was applied for 200 ms during the sound presentation and evaluated the animals' performance in the sound termination detection task (*Figure 1m*). The laser light was applied for 200 ms starting 200 ms after sound onset. The laser light activated PV+ cells in AAF, which in turn suppressed the activity of other AAF cells (*Figure 1n*). The laser presentation during the ongoing sound did not result in any significant change in the animals' performance (*Figure 1o, p*, Δ ODI (off–on) = −0.02 ± 0.12). These experiments demonstrate that offset responses in AAF are behaviorally relevant in the sound termination detection task.

## Larger offset responses correlate with better detection of sound termination

Previous studies illustrated how offset responses in the ACx of rats and cats strongly depend on the fall-time of a sound (*Qin et al., 2007*; *Takahashi et al., 2004*). We used a faster fall-ramp to evoke higher amplitude offset responses and asked whether the amplitude of offset responses and the

animal's ability to detect sound termination are correlated. We used a similar experimental paradigm as in *Figure 1*, where the animal had to detect the end of the 9 kHz tone played at 60 dB SPL with, this time, a fall-ramp of 10 or 0.01 ms. As expected from previous studies, fast fall-ramps (0.01 ms) lead to significantly higher offset responses than longer ones (10.0 ms), as tested during awake passive recordings (*Figure 2a*, *Figure 2—figure supplement 1*) or when animals were performing a behavioral task (*Figure 2b*). We confirmed that the sounds with a 0.01 ms fall-ramp are not causing an additional artificial onset response (*Figure 2—figure supplement 2*). Using an ultrasensitive microphone, we recorded acceleration traces of 9 kHz PT played at 60 dB SPL with 0.01 and 1 ms rise- and fall-ramp (*Figure 2—figure supplement 2a, b*). For the fast fall-ramp, we detected a weak spectral splatter present for less than 0.5 ms and covering the frequency range between 6 and 30 kHz (*Figure 2—figure supplement 2c, d*). We then investigated if this spectral splatter could trigger any significant onset response that would be mixed with the actual offset responses. First, we asked if neurons with a best frequency (BF) within the spectral splatter frequency range displayed a more significant response to sound termination. We did not observe any increased offset response in those neurons compared to neurons with other BF (*Figure 2—figure supplement 2e*), indicating that putative onset responses to the splatter could not be significant. Then, we asked if offset responses evoked by fast and slow ramps correlated. If offset responses were rather onset responses to the spectral splatter instead of actual offset responses, no correlation between these onset responses and the offset responses triggered by a slow ramp would be expected. We showed that the offset responses to fast and slow ramps correlated highly (*Figure 2—figure supplement 2f*, $\rho$ = 0.71, p < 0.0001, Spearman correlation), suggesting that our protocol allows us to identify actual offset responses. Finally, we asked if the responses at sound termination would influence the onset responses of a tone played shortly after. As offset and onset responses are driven by different sets of synapses (*Scholl et al., 2010*), we would expect the onset responses of the second sound to be affected by the first sound only if the response at sound termination were onset responses to the artifact, but not if they were actual offset responses. We found that onset responses were suppressed by preceding onset responses but not affected by preceding offset responses (*Figure 2—figure supplement 2g*), even when the interval was as short as 50 ms. This suggests that the offset responses we recorded are driven by different synapses that onset responses and cannot be onset responses.

The analysis of ODI showed that mice could correctly detect when sounds ended for tones with both short and long ramps and no significant difference in ODI between the two ramps were observed at the end of the test sessions (*Figure 2c–e*, Δ ODI (0.01–10 ms) = −0.01 ± 0.05). However, at the beginning of the test sessions, mice were better at detecting sound offset when the ramp was fast. This suggests that sounds terminated with a fast fall-ramp, triggering a more significant offset response, were possibly easier to detect at the beginning of the sessions when the task was still new and more difficult than after exposure to more trials. This result is in line with previous findings showing cortical involvement in challenging but not easy tasks (*Ceballo et al., 2019*; *Christensen et al., 2019*; *Dalmay et al., 2019*; *Kawai et al., 2015*). There was no significant difference in the reaction times for both tested ramps (*Figure 1—figure supplement 4*).

To confirm that more prominent offset responses help mice detecting sound termination, we performed another behavioral experiment with optogenetics, this time manipulating offset responses evoked by fast ramp (*Figure 2f*). As previously shown (*Figure 1d, e*), the laser significantly activated PV+ cells, resulting in the suppression of high amplitude offset responses in non-PV+ cells (*Figure 2g*). Minimizing high amplitude offset responses in AAF significantly decreased the performance to detect sound termination (*Figure 2h,i*, Δ ODI (off–on) = 0.25 ± 0.09). To control that the light itself, without ChR2, was not causing any changes in behavioral performance, we repeated the same experiments in wild-type animals (*Figure 2j*). The laser alone had no effect, neither at the neural (*Figure 2k*) nor at the behavior level (*Figure 2l and m*, Δ ODI (off–on) = −0.04 ± 0.09). Finally, we performed experiments in PV-ChR2 animals where the laser light was applied for 200 ms during the sound presentation (*Figure 2n*). The laser light activated PV+ cells in AAF, which in turn suppressed the activity of other AAF cells (*Figure 2o*). The laser presentation during the ongoing sound did not result in any significant change in the animals' performance (*Figure 2p, q*, Δ ODI (off–on) = 0.06 ± 0.14). Together, these experiments confirm that changing offset responses, but not sustained activity, in AAF influences behavioral performance. They demonstrate that the animal uses the neuronal activity following sound termination in AAF to detect sound termination.

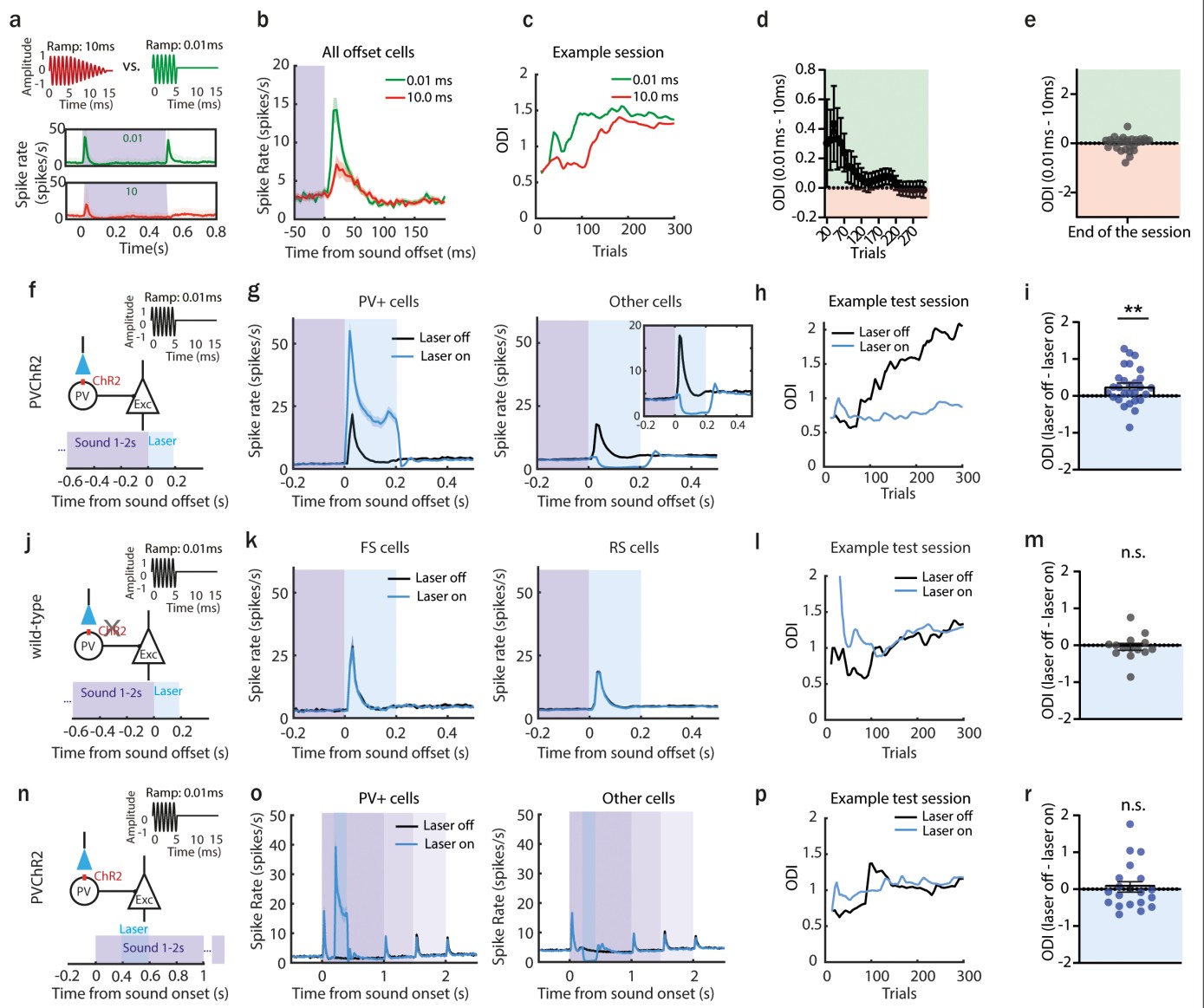

**Figure 2.** Larger offset responses correlate with better detection of sound termination. (**a**) Sounds used in the behavioral task (0.01 or 10 ms fall-ramp) and poststimulus time histogram (PSTH; mean ± standard deviation [STD]) of responses they evoked in acute anterior auditory field (AAF) recordings. (**b**) PSTH (mean ± standard error of mean [SEM]) averaged over AAF neurons (*n* = 169, 2 animals) during sound termination detection task (green line: 0.01 ms fall-ramp; red line: 10.0 ms fall-ramp). (**c**) Offset detection index (ODI) of example session during offset detection task. Green and red lines represent the ODI for short (0.01 ms) and long (10 ms) offset ramps. (**d**) The difference in ODI for sounds with short and long fall-ramps through the behavior session, *n* = 28 sessions, 10 animals. Green and red shaded areas indicate that offset detection was better for short or long ramps, respectively. (**e**) Comparison of ODI for 0.01 and 10.0 ms ramp at the end of the session (p > 0.99, *n* = 28 sessions, 10 animals, 1 sample Wilcoxon test). (**f**) Schematic of experimental design: a 9 kHz pure tone (PT) with 0.01 ms fall-ramp was used, and the laser was applied for 200 ms following sound termination in animals expressing channelrhodopsin-2 (ChR2) in parvalbumin-positive (PV+) cells. (**g**) The activity of PV+ cells (mean ± SEM) following sound termination in laser-on (blue) and laser-off (black) trials (left), *n* = 336 cells, 28 sessions, 6 animals. The activity of other cells (mean ± SEM) following sound termination in laser-on (blue) and laser-off (black) trials (right), *n* = 2249 cells, 28 sessions, 6 animals. (**h**) ODI within an example session during offset detection task. Blue and black lines represent ODI for laser-on and laser-off trials, respectively. (**i**) Difference in ODI for laser-off and laser-on trials at the end of the training (mean ± SEM), **p = 0.0095, *n* = 28 sessions, *n* = 6 animals, 1 sample Wilcoxon test. (**j**) Schematic of control experiment: a sound of 9 kHz with 0.01 ms fall-ramp was used, and the laser was applied for 200 ms following sound termination in wild-type animals. (**k**) Population activity (mean ± SEM) following sound termination in laser-on (blue) and laser-off (black) trials in fast (left), *n* = 103 cells, 14 sessions, 3 animals, and regular spiking neurons (right), *n* = 930 cells, 14 sessions, 3 animals. (**l**) ODI for an example session during offset detection task. Blue and black lines represent ODI for laser-on and laser-off trials, respectively. (**m**) Difference in ODI for control animals in laser-off and laser-on trials at the end of the training (mean ± SEM), p = 0.41, *n* = 14 sessions, 3 animals, 1 sample Wilcoxon test. (**n**) Schematic of the experiment with laser suppression during the sound presentation: a 9 kHz PT with 0.01 ms fall-ramp was played, and the laser was applied for 200 ms following sound onset in animals expressing

*Figure 2 continued on next page*

*Figure 2 continued*

ChR2 in PV+ cells. (**o**) Population activity (mean ± SEM) in laser-on (blue) and laser-off (black) trials of PV+ cells (left), *n* = 269 cells, 20 sessions, 4 animals, and other cells (right), *n* = 1414 cells, 20 sessions, 4 animals. (**p**) ODI for an example session during offset detection task. Blue and black lines represent the ODI for laser-on and laser-off trials, respectively. (**r**) Difference in ODI for laser-off and laser-on trials at the end of the training (mean ± SEM), p = 0.74, *n* = 20 sessions, 4 animals, 1 sample Wilcoxon test.

The online version of this article includes the following figure supplement(s) for figure 2:

**Figure supplement 1.** Offset responses evoked by sounds terminated with different fall-ramps emerge already in medial geniculate body (MGB).

**Figure supplement 2.** Properties of 9 kHz pure tones played at 60 dB SPL with different rise- and fall-ramps.

## The activity of AAF neurons in a sound termination detection task can be predictive of the animal's performance

As the suppression of auditory offset responses in AAF affects performance, we asked if AAF activity during a single trial could be predictive of the animal's behavior. We used a logistic regression model to predict the mouse's action from the single-trial population activity (cross-validated, L2 penalty, see methods). We examined the classifier accuracy for the model trained and tested on the spontaneous, onset, sustained, offset, and late response (*Figure 3a*) from an equal number of hit and miss trials from all experiments with fast ramp. The decoding performed on the 'late' phase served as a control, confirming that specific patterns of activity within individual neurons reflect action of the animal, hence leading to better decoding. We compared the classifier accuracy trained on different response types and found that offset and late responses allowed for significantly better action decoding than spontaneous or sustained responses (*Figure 3c*, Spont.: 59.8% ± 1.4%, Onset: 63.1% ± 1.5%, Sustained: 60.8% ± 1.5%, Offset: 66.4% ± 1.5%, Late: 76.7% ± 1.2%). These results suggest that AAF offset responses can be informative on the animal's decision emphasizing the behavioral relevance of AAF offset responses. However, the classifier accuracy was not significantly different between the offset and onset windows, suggesting that perceiving a sound and its termination are intricately linked.

## AAF is highly specialized for processing information on sound termination

Knowing that offset responses in AAF are behaviorally relevant and influence sound termination perception, we next asked what mechanisms drive these cortical offset responses and what properties distinguish them from subcortical ones. We performed awake electrophysiological recordings in AAF and MGB and analyzed the response profile dynamics of cells within both regions. We recorded multiunit activity evoked by 50 ms PT with varying frequency (4–48.5 kHz) and sound level (0–80 dB SPL) presented with randomized interstimulus intervals (ISIs) (500–1000 ms).

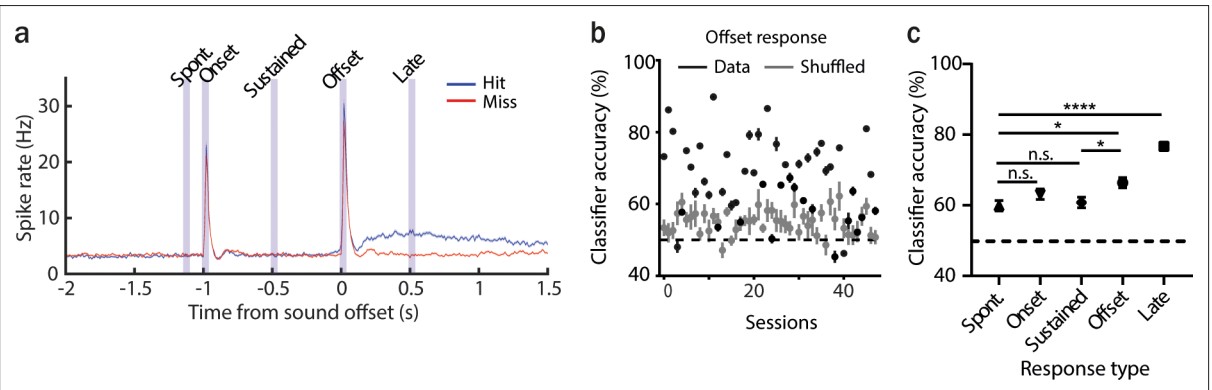

**Figure 3.** The activity of anterior auditory field (AAF) neurons can be predictive of animal performance. (**a**) Averaged poststimulus time histogram (PSTH; mean ± standard error of mean [SEM]) of AAF neuron responses to 1 s long pure tone (PT; 9 kHz) played at 60 dB SPL during hit (blue) and miss (red) trials, *n* = 5076 cells, 59 sessions, 12 animals. (**b**) Classification accuracy based on offset responses (mean ± SEM) for real and shuffled data. (**c**) Comparison of classifier accuracy of decoders trained and tested on spontaneous activity, onset, sustained, offset, and the late response of AAF neurons (mean ± SEM): spont. vs offset: p = 0.027; offset vs sustained: *p = 0.037; spont. vs. late: ****p < 0.0001, *n* = 48 sessions, 12 animals, Friedman test with multiple comparisons.

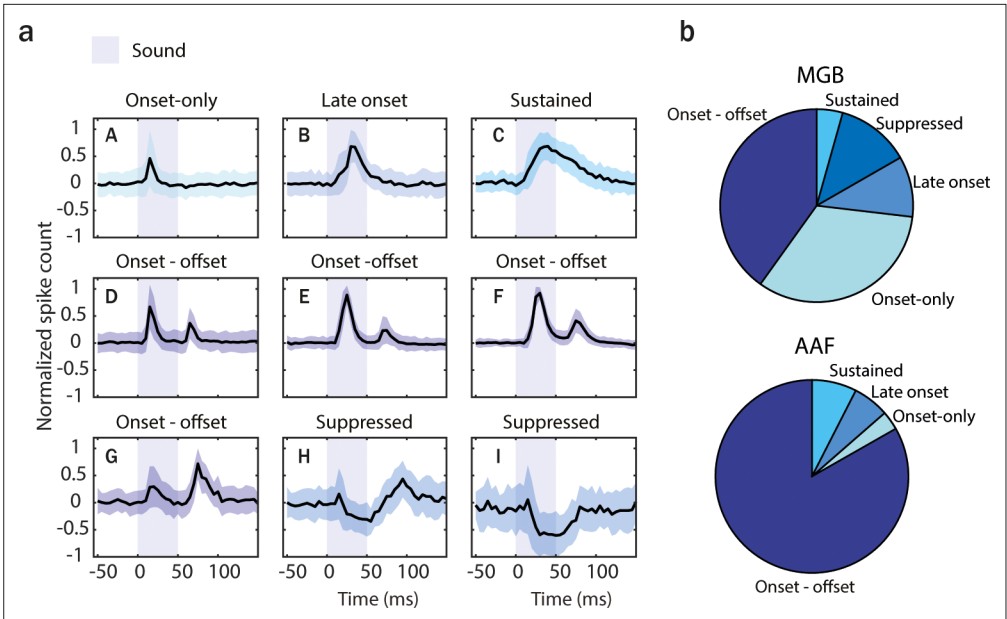

**Figure 4.** Anterior auditory field (AAF) has significantly more offset-responsive neurons than its input nucleus. (**a**) Results of *k*-means clustering performed on both medial geniculate body (MGB; *n* = 779, 6 animals) and AAF (*n* = 346, 6 animals) neuron's responses (time window: 25–75 ms, bin size: 5 ms) evoked by 50 ms pure tone (PT) with varied frequency (4–48.5 kHz) and sound level (0–80 dB SPL) presented with randomized interstimulus interval (ISI; 500–1000 ms). Graphs represent the mean signal of cells belonging to each cluster. Data represent mean ± STD. The blue shaded bars represent the tone. (**b**) Representation of cells with distinct temporal dynamic of responses in MGB (onset-only [**A**]: 40.1%, late-onset [**B**]: 10.2%, sustained [**C**]: 4.4%, onset–offset [**D-G**]: 33.0%, and suppressed [**H-I**]: 12.3%) and AAF (onset-only [A]: 3.2%, late-onset [B]: 6.1%, sustained [C]: 7.5%, and onset–offset [D-G]: 83.2%). See also *Figure 4—figure supplements 1 and 2*.

The online version of this article includes the following figure supplement(s) for figure 4:

**Figure supplement 1.** Davies–Bouldin index for a different number of clusters.

**Figure supplement 2.** 2D representation of temporal dynamics of medial geniculate body (MGB) cells recorded during 4 experiments with a 64-channel electrode (16 × 4).

*K*-Means clustering of spike-sorted unit (SU) activity was used to identify cells with distinct temporal dynamics. The clustering method was performed on the averaged poststimulus time histogram (PSTH) in both MGB (*n* = 779 SU, 6 animals) and AAF (*n* = 346 SU, 6 animals) recordings pooled together. The analysis time window for the clustering was chosen to emphasize the offset rather than the onset responses (25–75 ms, bin size: 5 ms). Davies–Bouldin evaluation was used to determine the optimal number of clusters (*Figure 4—figure supplement 1*). Nine clusters were identified, reflecting five main temporal categories of auditory evoked responses: *onset-only*, *late-onset*, *onset–offset*, *sustained*, and *suppressed* (*Figure 4*). Few clusters with the same temporal dynamic pattern were detected (e.g., D–G) resulting from various latencies, width, and the ratio of offset/onset responses. These clusters were merged for further analysis. In MGB, both *onset-only* and *onset–offset* cells represented the most prominent clusters: 40.1% and 33.0%, respectively (*Figure 4b*). Cells with these two temporal response patterns also revealed separate anatomical clusters within MGB (*Figure 4—figure supplement 2*; *He, 2002*). *Suppressed* responses were found in 12.3%, *late-onset* responses in 10.2%, and *sustained* in 4.4% of cells. In AAF, most cells were *onset–offset* responsive (83.2%). Other categories were represented in much smaller proportions: *onset-only* (3.2%), *late-onset* (6.1%), and *sustained* (7.5%). In contrast to MGB, no *suppressed* cells were recorded in AAF. The overrepresentation of onset–offset responses in AAF compared to the preceding nucleus of the auditory pathway indicates that AAF is highly specialized in processing information on sound termination.

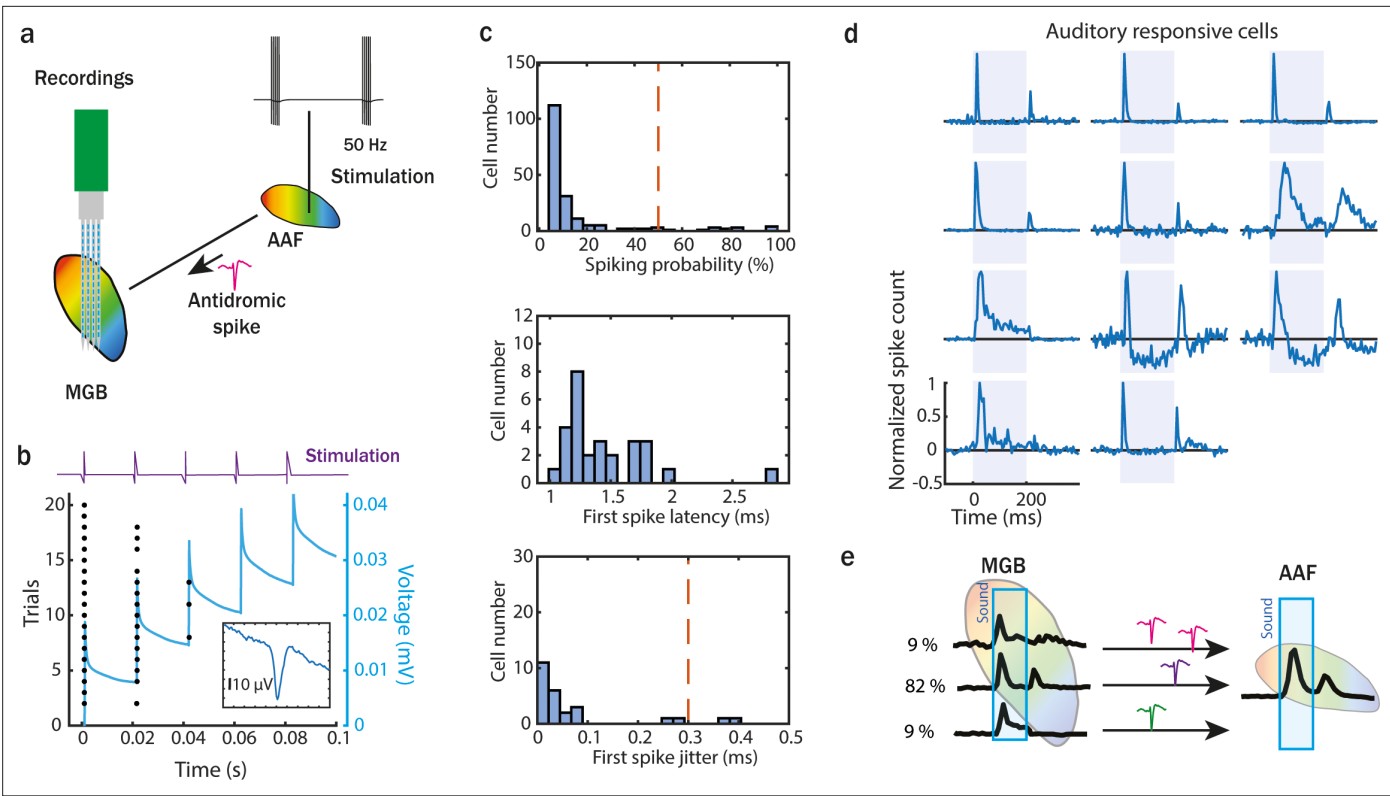

**Figure 5.** Onset–offset-responsive cells are the main inputs from medial geniculate body (MGB) to anterior auditory field (AAF). (**a**) Illustration of experimental setup to perform an antidromic stimulation of MGB neurons projecting to AAF. The stimulation tip was placed in AAF after field identification (based on functional tonotopy). Monophasic electric pulses were delivered with 50 Hz at 30 µA. A 64-channel electrode was inserted in MGB to record antidromic activity. (**b**) Example MBG unit spike raster of antidromic activity for 20 trials (5 pulses in each trial). The blue line represents the increasing current injected during electric stimulation. (**c**) The number of recorded antidromic spikes in MGB neurons, where at least one spike following stimulation was detected in the time window from 1 to 5 ms after stimulation (top). Latency (middle) and jitter (bottom) of first antidromic spikes in MGB cells which were detected in at least half of the electric stimulation trials. (**d**) Poststimulus time histogram (PSTH) of auditory responsive MGB cells projecting to AAF identified during the antidromic experiment. Three main temporal categories of auditory evoked responses were identified: onset–offset, sustained and onset. MGB cells were considered AAF input if (1) antidromic spikes were detected in more than 50% of trials and (2) antidromic spikes jitter was lower than 0.3 ms. (**e**) Illustration of temporal dynamic and proportion of MGB cells projecting to AAF.

The online version of this article includes the following figure supplement(s) for figure 5:

**Figure supplement 1.** Response dynamics of medial geniculate body (MGB) cells recorded during antidromic stimulation experiments.

## Onset–offset-responsive cells are the main inputs from MGB to AAF

As AAF contains cells with mainly onset–offset responses – unlike MGB or A1 (*Sołyga and Barkat, 2019*), we asked whether these cortical offset responses were inherited from MGB cells or whether they arose de novo in AAF. We combined in vivo electrophysiological recordings in MGB (n = 1548 SU, 5 animals) with antidromic stimulation of AAF, followed by PT stimulation to characterize the temporal dynamics of cells projecting from MGB to AAF (*Figure 5a, Figure 5—figure supplement 1*). A stimulating electrode was inserted into the previously functionally identified AAF (see methods). Pulse trains of monophasic square pulses were used for the electric stimulation (*Figure 5b*). We identified MGB cells directly connected to AAF by analyzing the percentage and latency of responses to the first stimulation pulse in each train. MGB cells that fired as a response to AAF stimulation in at least 50% of the trials with a first spike latency of 1–3 ms and a trial-to-trial latency jitter lower than 0.3 ms (*Serkov et al., 1976*) were considered to be sending direct inputs to AAF (*Figure 5c*). These MGB cells were clustered mainly as onset–offset cells (82%). We also identified some sustained (9%) and onset (9%) cells projecting from MGB to AAF, but their representation was significantly lower (*Figure 5d, e*). These results indicate that offset responses in AAF are mainly inherited from MGB. Whether further processing of these offset responses within the cortex took place was, however, still unclear.

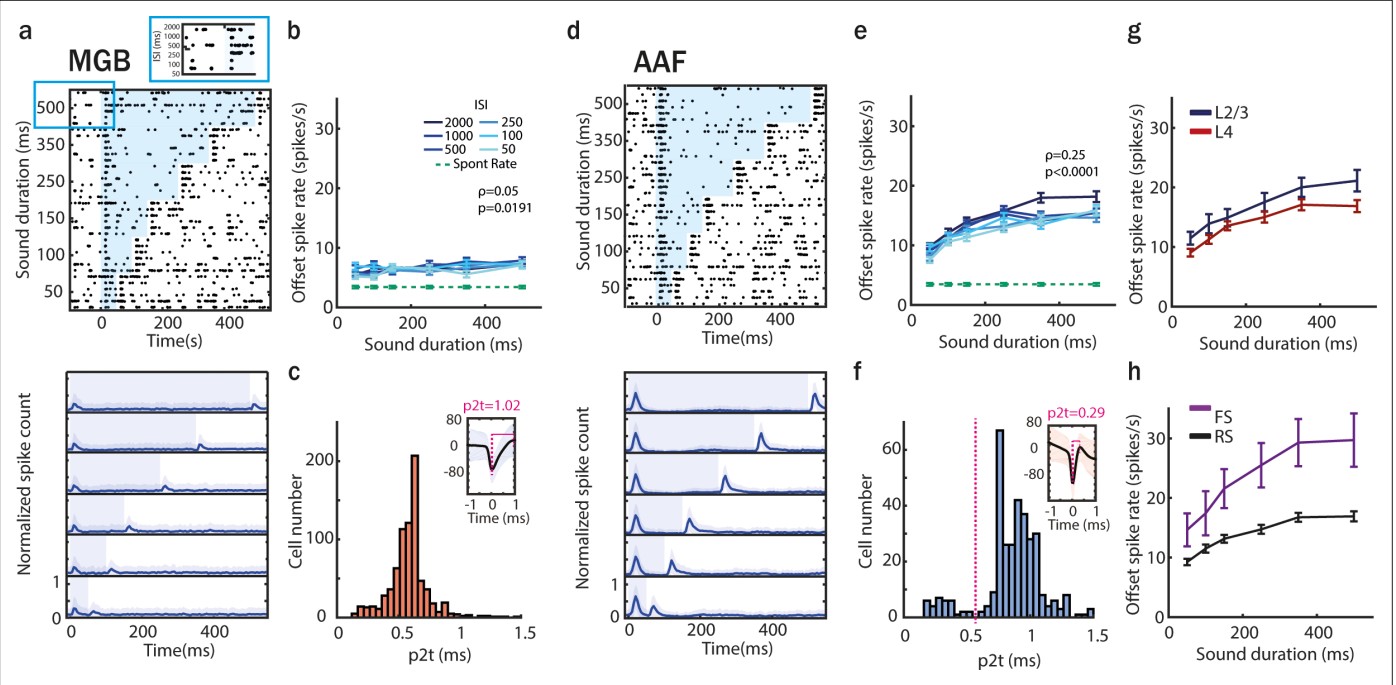

**Figure 6.** Offset responses increase with sound duration mainly in anterior auditory field (AAF). (**a**) Raster plot of an example medial geniculate body (MGB) neuron's response to pure tone (PT; frequency chosen based on offset tuning) with sound duration varying between 50 and 500 ms and interstimulus interval (ISI) between 50 and 2000 ms (top) and poststimulus time histogram (PSTH; mean ± STD) averaged over all neurons population (bottom). The blue shaded bars represent the tone. (**b**) MGB neurons offset responses (mean ± standard error of mean [SEM]) to PT with increasing duration across all ISI of 2000 ms (correlation between sound duration and response rate: PT, $\rho$ = 0.05, p = 0.019, n = 307 SU, 6 animals, Spearman correlation). Comparison of offset responses: 50 vs 100 ms: n.s. p = 0.080; 50 vs 150 ms: n.s. p = 1; 50 vs 250 ms: p = 0.032; 50 vs 350 ms: n.s. p = 0.70; 50 vs 500 ms: n.s. p = 0.91, Dunn's multiple comparisons test. (**c**) Distribution of peak-to-trough times (p2t) of MGB neurons. (**d**) Raster plot of an example AAF neuron's response to PT (9 kHz) played at 60 dB SPL with sound duration varying between 50 and 500 ms and ISI between 500 and 2000 ms (top), and PSTH (mean ± STD) averaged over all neurons population (bottom). The blue shaded bars represent the tone. (**e**) AAF neurons offset responses (mean ± SEM) to PT with increasing duration across ISI of 2000 ms (correlation between sound duration and response rate: PT, $\rho$ = 0.25, p < 0.0001, n = 285 SU, 6 animals, Spearman correlation). Comparison of offset responses: 50 vs 100 ms: n.s. p = 0.57; 50 vs 150 ms: n.s. p > 0.99; 50 vs 250 ms: n.s. p = 0.082; 50 vs 350 ms: p = 0.041; 50 vs 500 ms: p < 0.0001, Dunn's multiple comparisons test. (**f**) Distribution of p2t of AAF neurons. (**g**) Comparison of offset spike rate in L2/3 and L4 neurons in AAF for sounds with duration varying between 50 and 500 ms and longest tested ISI of 2000 ms (mean ± SEM), L2/3: $\rho$ = 0.26, p < 0.0001, $n_{2/3}$ = 87, L4: $\rho$ = 0.24, p < 0.0001, $n_4$ = 198, Spearman correlation. (**h**) Comparison of offset spike rate of fast spiking and regular spiking (RS) neurons in AAF for sounds with duration varying between 50 and 500 ms and longest tested ISI of 2000 ms (mean ± SEM), FS: $\rho$ = 0.29, p = 0.0001, $n_{FS}$ = 28, RS: $\rho$ = 0.24, p < 0.0001, $n_{RS}$ = 257, Spearman correlation.

## Offset responses increase with sound duration mainly in AAF

Given the presence of offset responses in MGB and AAF, we next asked whether their properties were similar in both regions. To reveal differences in offset processing, we decided to check how offset responses in MGB and AAF were affected by different sound properties such as sound duration, spectral content, or temporal complexity. We first addressed the dependence of offset responses on sound duration (*Scholl et al., 2010*; *Sołyga and Barkat, 2019*). We recorded responses in MGB and AAF offset cells (clustering based on *Figure 4*) to 60 dB SPL tones with durations varying between 50 and 500 ms and ISIs varying between 50 and 2000 ms (*Figure 6a, d*). For MGB, the tone frequency was dependent on the offset BF of neurons in each session (because of narrow offset tuning of MGB neurons); for AAF, a fixed PT of 9 kHz was used (as most of the widely tuned cells were responding to this frequency). The correlation between sound duration and offset spike rate evoked by tones in onset–offset cells was very weak in MGB but much stonger in AAF (*Figure 6b, e*). In MGB, population responses showed almost no difference in offset spike rate evoked by tones when durations changed between 50 and 500 ms (*Figure 6b*). In AAF, however, differences in tone duration were significantly reflected by increased offset spike rates for the longest sounds (*Figure 6e*).

To explore whether the dependence of offset responses on sound duration was a result of AAF computations in layer 2/3 (L2/3) or was already present in the input layer 4 (L4), we compared the

dependence of offset responses on sound duration in these layers (*Figure 6g*). Our recordings spanned the range of 150–600 µm from the pia surface, corresponding mainly to L2/3 (150–300 µm) and L4 (300–500 µm) (*Meng et al., 2017*). A significant correlation was present both in L2/3 and L4, suggesting that the increase in offset response amplitude with sound duration is not unique to one layer.

As MGB does not contain fast-spiking (FS) interneurons (*Bartlett, 2013*), we then asked whether their presence in AAF could be driving the dependence of offset responses on sound duration in this cortical region (*Figure 6c, f*). We distinguished putative FS and RS neurons based on the peak-to-trough times (p2t) of their spike waveforms (*Figure 6f*). FS units were defined as having a p2t smaller than the minimum between the two peaks of the p2t distribution (0.55 ms), in accordance with previous studies (*Moore and Wehr, 2013*). The unimodal distribution of p2t in MGB confirmed the lack of FS in this region (*Figure 6c*). We found a significant correlation between offset spike rate and sound duration in FS and RS neurons (*Figure 6h*), ruling out the possibility that one of these cell populations is alone driving the dependence of offset responses on sound duration in AAF.

Together, these results reveal an essential difference between AAF and MGB offset encoding and demonstrate a clear amplification of the dependence of offset responses on sound duration in AAF as compared to MGB.

## Offset responses to WN stimulation are present in AAF but not in MGB

Next, we compared offset responses in MGB and AAF evoked by sounds with different spectral complexity. We recorded responses in both regions to 500 ms PT, and WN bursts played at 60 dB SPL (for MGB, the PT frequency was chosen based on offset BF of neurons in each session; for AAF, a fixed PT of 9 kHz was used). Our results showed very distinct neuronal activity patterns in response to WN or the spectrally less complex PT, both in MGB and in AAF (*Figure 7*). In MGB, 500 ms WN evoked no offset responses above spontaneous activity, unlike PT of the same length (*Figure 7a, c*). In AAF, however, both PT and WN did evoke offset responses (*Figure 7b, d*). The lack of offset responses evoked by WN in MGB, and their significant presence in AAF, revealed offset responses generated de novo in the cortex.

We then asked whether offset responses to WN stimulation differed between different neuronal populations of AAF. We found that responses to WN were significantly more sustained in L4 than in L2/3 (*Figure 7e*, calculated in a window of 100–500 ms following sound onset), despite having similar offset responses in both layers (*Figure 7f*). It seemed that responses to WN stimulation, even if not present among MGB inputs, arose already in AAF input layer. Next, we compared offset responses evoked by PT and WN in putative FS and regular spiking (RS) neurons (*Figure 7g, h*). As expected, offset responses were more prominent and faster in FS than RS neurons following PT termination (*Figure 7l*, *Figure 7—figure supplement 1a, b*). However, there was not a significant difference in spike rate and latency between FS and RS neurons following WN termination (*Figure 7—figure supplement 1c*). The comparison of ratios of offset/onset responses between cell and sound types showed that the ratio for FS neurons was significantly smaller than the ratio for RS neurons with WN stimulation (*Figure 7i*). This relative decrease of offset responses in FS neurons could reduce inhibition and result in enhanced activity of excitatory cells, leading to offset responses generated de novo in AAF. These findings suggest that FS neurons could play an important role in the cortical processing of WN offset responses.

The PSTHs of AAF neurons responding to PT and WN stimulation indicated that WN evoked more sustained activity than PT (*Figure 7b*). More specifically, WN gave rise to sharp onset responses followed by suppression, and then an activity rebound at around 200 ms followed by a second suppression phase and another rebound. Could this specific response of neurons to WN stimulation influence the strength of AAF offset responses? Comparing the firing rate of FS and RS cells in AAF 50 ms before the sound termination revealed that they were significantly stronger for WN than PT stimulation (*Figure 7j*). This extended firing could affect the generation of offset for WN stimulation. Offset responses evoked by WN were not increasing as a function of sound duration in neither MGB nor AAF (*Figure 7—figure supplement 2*).

Studying offset responses evoked by WN bursts revealed two fascinating differences between MGB and AAF processing. First, offset responses to WN stimulation are not present in MGB but are

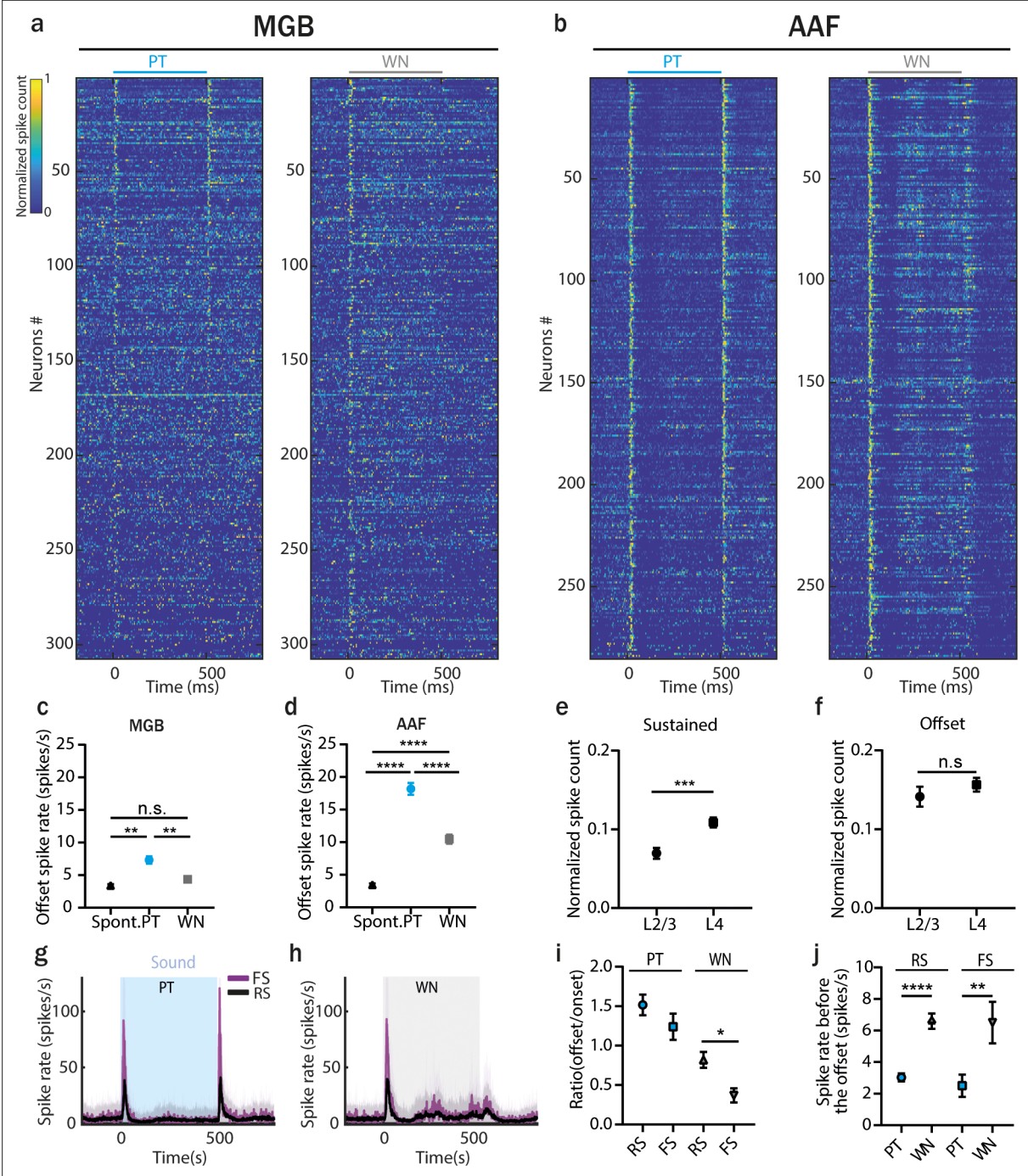

**Figure 7.** Offset responses to white noise (WN) stimulation are present in anterior auditory field (AAF) but not in medial geniculate body (MGB). (**a**) Normalized poststimulus time histogram (PSTH) of MGB neurons to 500 ms pure tone (PT) or WN bursts, bin size: 5 ms. Data are sorted by descending spike rate at the PT offset. (**b**) Normalized PSTH of AAF neurons to 500 ms PT or WN bursts, bin size: 5 ms. Data are sorted by descending spike rate at the PT offset. (**c**) Comparison of MGB offset responses evoked by PT and WN for onset–offset cells. Data represent mean ± standard error of mean (SEM), PT vs spont. rate: **p = 0.0079; WN vs spont. rate: p > 0.99, PT vs WN: **p = 0.0037, $n$ = 307, Friedman test with multiple comparisons. (**d**) Comparison of AAF offset responses evoked by PT and WN for onset–offset cells. Data represent mean ± SEM, PT vs spont. rate: ****p < 0.0001; WN vs spont. rate: ****p < 0.0001, PT vs WN: ****p < 0.0001, $n$ = 285, Friedman test with multiple comparisons. (**e**) Comparison of sustained responses (calculated in the window: 100–500 ms) for AAF cells from L2/3 and L4 evoked by 500 ms WN stimulation (mean ± SEM), ***p = 0.0003, $n_{2/3}$ = 87, $n_4$ = 198, Mann–Whitney test. (**f**) Comparison of offset responses for AAF cells from L2/3 and L4 evoked by 500 ms WN stimulation (mean ± SEM), p = 0.30, $n_{2/3}$ = 87, $n_4$ = 198, Mann–Whitney test. (**g, h**) Averaged PSTH of fast ($n$ = 29) and regular ($n$ = 249) spiking AAF neuron's response to PT and WN bursts played at 60 dB SPL with sound duration 500 ms and interstimulus interval (ISI) between 500 and 2000 ms. (**i**) Ratio of offset/onset responses evoked by

*Figure 7 continued on next page*

*Figure 7 continued*

500 ms PT or WN in fast spiking and regular spiking (RS) AAF neurons (mean ± SEM), p = 0.020, $n_{FS}$ = 28, $n_{RS}$ = 246, Mann–Whitney test. (**j**) Spike rate preceding sound offset (calculated in window: 450–500 ms) in AAF neurons for longest ISI of 2000 ms (mean ± SEM), RS: ****p < 0.0001, n = 257; FS: **p = 0.0013, n = 28, Wilcoxon test.

The online version of this article includes the following figure supplement(s) for figure 7:

**Figure supplement 1.** Offset processing in fast spiking (FS) and regular spiking (RS) anterior auditory field (AAF) neurons.

**Figure supplement 2.** Offset responses to white noise (WN) stimulation do not increase with sound duration.

present in AAF. Second, cells in AAF seem to follow ongoing WN stimulation with a bursting activity happening every ~200 ms.

## Offset responses encode more than just silence

The precise detection of fast changes in sound frequency and level is crucial for gap detection and vocalization (**Kopp-Scheinpflug et al., 2018**; **Sollini et al., 2018**). We asked if offset responses in MGB and AAF could encode more than silence – that is, sound termination – such as important changes within temporally discontinuous sounds (**Lu et al., 2001**). We recorded responses to a multi-frequency component sound in MGB (n = 275 SU, 6 animals) and AAF (n = 284 SU, 6 animals). The complex sound consisted of three frequency components (20, 14, and 9 kHz) played at 60 dB SPL, which had a common onset but ended at different time points (300, 400, and 500 ms). The offset responses

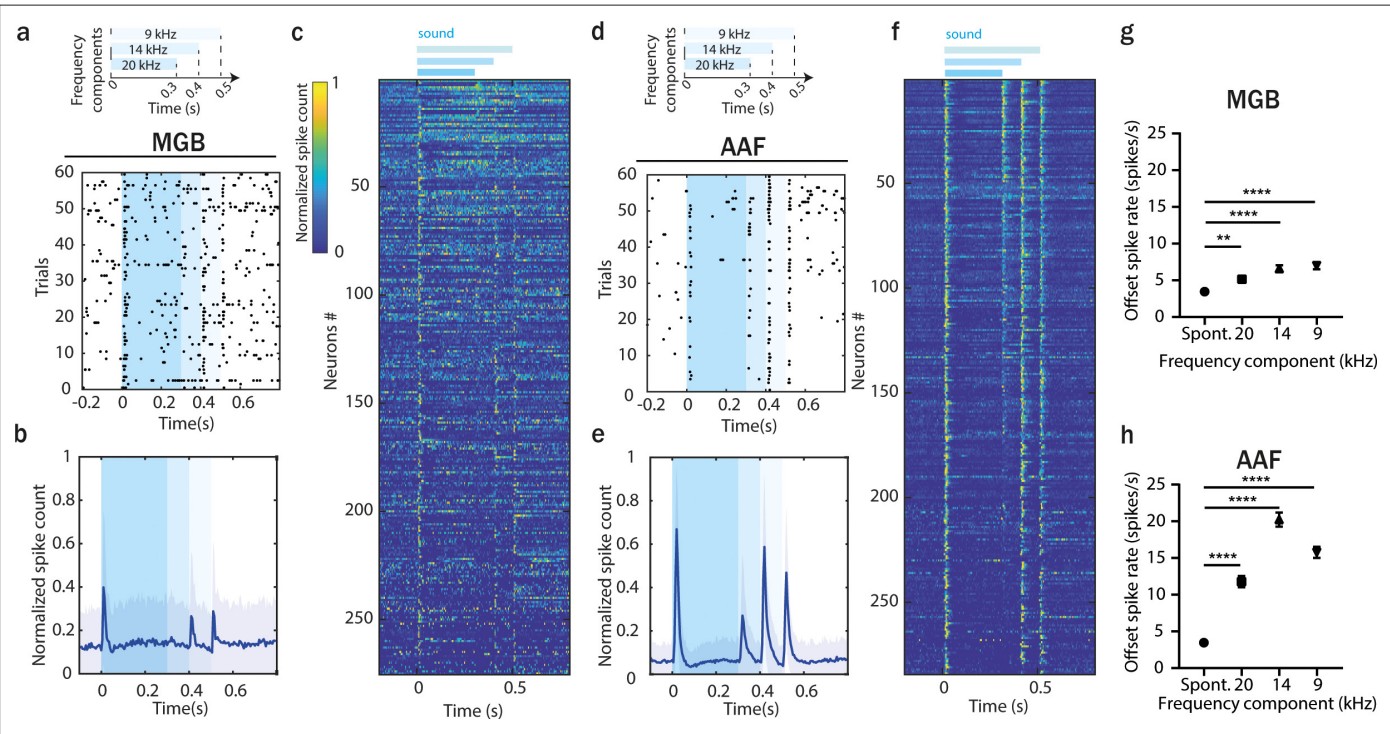

**Figure 8.** Offset responses encode more than just silence. (**a**) Raster plot of an example MGB neuron's response to three-component stimuli. (**b**) Averaged poststimulus time histogram (PSTH; mean ± STD) of MGB neuron's response to three-component stimuli, n = 275, n = 6 animals. (**c**) Normalized PSTH of MGB neuron's response to three-component stimuli, bin size: 5 ms. Data are sorted by descending spike rate at the first component termination. (**d**) Raster plot of an example AAF neuron's response to three-component stimuli. (**e**) Averaged PSTH (mean ± STD) of AAF neuron's response to three-component stimuli, n = 284, n = 6 animals. (**f**) Normalized PSTH of AAF neuron's response to three-component stimuli, bin size: 5 ms. (**g**) Comparison of spiking rate of MGB neurons following removal of each frequency component. Data represent mean ± standard error of mean (SEM), Spont. vs 20 kHz: **p = 0.0011, Spont. vs 14 kHz: ****p < 0.0001, Spont. vs 9 kHz: ****p < 0.0001, n = 275, n = 6 animals, Friedman test with multiple comparisons. (**h**) Comparison of spiking rate of AAF neurons following removal of each frequency component. Data represent mean ± SEM, ****p < 0.0001, n = 284, n = 6 animals, Friedman test with multiple comparisons.

The online version of this article includes the following figure supplement(s) for figure 8:

**Figure supplement 1.** Distinct spectral and temporal tuning properties of medial geniculate body (MGB) and anterior auditory field (AAF) cells.

evoked by removing one or two frequency components demonstrate that neurons can encode the disappearance of a frequency component in an ongoing sound, especially in AAF (*Figure 8a–f*). A single MGB neuron usually encoded the removal of one or two frequency components. In contrast, most AAF neurons encoded the removal of all frequency components. However, at the population level, the termination of all three components was significantly encoded in both MGB and AAF activities (*Figure 8g, h*). Interestingly, removing the first frequency component (20 kHz) evoked the most negligible offset response, which, in MGB, was close to spontaneous activity. Removal of the last component (9 kHz), followed by silence, evoked the highest offset response within MGB neurons. In contrast, the highest offset response was present in AAF after the removal of the second component (14 kHz). AAF neurons seem to have a stronger ability to integrate information over spectrally and temporally complex ongoing sounds and not only to respond to silence. This ability to encode important changes within continuous sound could be crucial for processing temporally discontinuous sounds, making AAF an interesting field to study the encoding of vocal calls.

## Discussion

As the auditory system very robustly represents timing information, it is a model of choice to study neural offset responses evoked by the disappearance of a stimulus. In this study, we show that minimizing AAF offset responses decreases the mouse performance to detect sound termination, thus revealing their importance at the behavioral level. By combining in vivo electrophysiology recordings in AAF and MGB of awake mice, we also demonstrate that AAF inherits, amplifies, and sometimes even generates de novo offset responses. These results are of high importance for all studies on sensory processing, as the mechanisms determining specificities in cortical vs thalamic processing revealed by our studies could be shared between the different sensory areas.

The functional significance of offset responses was long under debate (*Saha et al., 2017*). Here, we show that minimizing AAF offset responses significantly decreases the performance of mice to detect sound termination (*Figures 1 and 2*). In addition, sounds terminating with a fast fall-ramp and triggering a more significant offset response seems to be easier to detect at the beginning of a behavioral session when the task is still new and more challenging than after exposure to more trials. A possible explanation for this dependence on task difficulty could be related to the distinct involvement of the ACx during more or less challenging tasks. Previous studies have shown that the cortex could be required for challenging tasks, but less so for easier tasks or when the task is learned well (*Ceballo et al., 2019*; *Christensen et al., 2019*; *Dalmay et al., 2019*; *Kawai et al., 2015*). When the task is easy or familiar, high and low amplitude offset responses seem to be informative about sound termination to a similar extent. We also used logistic regression to show that AAF activity during a single trial could be predictive of the animal's performance (*Figure 3*). The neural activity during hit trials in AAF could be influenced by the animal's general motivation (*Fritz et al., 2003*), motor-related inputs (*Schneider, 2020*), or reward expectations (*De Franceschi and Barkat, 2021*).

Another approach to understanding the behavioral role of auditory offset responses could be to study where AAF is projecting and what could be the use of offset responses in these areas. It was recently shown that A1 transient offset responses to noise stimulation underlie the perception of sound duration (*Li et al., 2021*). As AAF supplies ~45% of the cortical input to A1 (*Lee and Winer, 2008*) and the modulation of acoustic information between A1 and AAF in the cat's ACx was shown to be dominated by a unidirectional AAF to A1 pathway (*Carrasco and Lomber, 2009*), it would be interesting to understand if A1 offset responses originate from AAF. Very recently, Bondanelli et al. argued for a role of recurrent A1 connectivity in shaping offset responses in cortex, including the fact that the offset response carries information about stimulus type (*Bondanelli et al., 2021*). The high level of information exchange between A1 and AAF raises the question of whether these mechanisms in A1 depend on AAF activity. It was also previously shown that offset responses in the secondary ACx are plastic and enhance the representation of a newly acquired, behaviorally relevant sound category (*Chong et al., 2020*). Whether the activity of AAF neurons is crucial for this plasticity to happen remains to be elucidated. Only recently, Nakata et al. revealed direct AAF connections to the secondary motor cortex, the primary somatosensory cortex, the insular ACx, and the posterior parietal cortex (*Nakata et al., 2020*). What role auditory offset responses in these fields play and whether they provide any information for association with the somatomotor system remain unanswered.

Spectrotemporal tuning properties of auditory neurons differ during the presentation of natural and synthetic stimuli (*David et al., 2007*; *Theunissen et al., 2001*). Natural sounds also commonly start abruptly, but their termination is obscured by sound reverberations and therefore stops less sharply than synthetic stimuli. In many of our experiments, we used a short 0.01 ms sinusoidal fall-ramp while keeping the rise-ramp at 4 ms. Our quantification of the spectral splatter (*Figure 2—figure supplement 2*) confirms that the neuronal responses measured at sound termination are not affected by the weak and short spectral splatter caused by a fast fall-ramp. In the future, the role of offset responses in detecting sound termination should be studied in natural environments, using, for example, vocalization calls (*Chong et al., 2020*). This would elucidate whether the auditory system, and MGB or AAF more specifically, evolved to meet the challenges of detecting naturally terminating sounds.

Our results demonstrate that cortical offset responses are not only inherited from the periphery (*Figures 4 and 5*) but also amplified by sounds with longer duration (*Figure 6*). Could the presence of FS interneurons in AAF, but not in MGB (*Bartlett, 2013*), be driving the dependence of offset responses on sound duration in this cortical region? The strength of the correlation between offset responses and sound duration was similar between FS and RS cells in AAF (*Figure 6*). However, FS neurons showed a significantly larger amplitude of offset responses in comparison to RS neurons. AAF was previously shown to have more PV+ cells than the other auditory primary region A1 (*Reinhard et al., 2019*). Mechanistically, we suggest that such a prominent PV network in AAF and the strong offset responses they exhibit could be crucial for evoking duration-dependent offset responses in AAF but not in A1 (*Sołyga and Barkat, 2019*) or MGB (*Figure 6*). With our antidromic experiments, we also showed that some of AAF cells receive inputs from sustained MGB cells (*Figure 5*). Whether or not the inputs from these cells are involved in the duration dependence of offset response in AAF should be explored further.

What could be the role of the cortex in tracking subtle differences in sound duration? At the behavioral level, one could speculate that the amplitude of offset responses would be needed to better track subtle differences in sound duration, especially for sounds shorter than 500 ms, covering a spectrum of most mouse calls (*Geissler and Ehret, 2002*). Additionally, the lack of increase in offset response amplitude with the duration of WN bursts (*Figure 7—figure supplement 2*) suggests that a stimulus has to contain a spectral structure to evoke duration-dependent offset responses (*Li et al., 2021*). Nevertheless, if the increase in the amplitude of offset responses with sound duration is a carrier of useful information or just a result of cortical cells being unable to handle short sounds remains unclear.

The spectral complexity of sounds is significantly modulating offset responses in the central auditory system in several ways. First, the lack of offset responses to WN in MGB onset–offset cells and their presence in AAF reveals offset responses generated de novo in the cortex (*Figure 7*). How could the differences in offset responses to WN stimulation be explained? The large spectral integration of thalamic inputs (*Figure 8—figure supplement 1*) in individual cortical neurons (*Liu et al., 2007*; *Vasquez-Lopez et al., 2017*) should not play a role, as no firing upon WN termination was observed in the thalamus. Interestingly, offset responses to WN stimulation were present in AAF both in L4 and L2/3, suggesting that it is a general property of AAF network to generate offsets de novo and not only an exclusive property of the superficial layer. What mechanisms could drive offset responses to WN in AAF? FS and RS neurons exhibit similar offsets responses to WN stimulation, thus not allowing us to speculate on their particular involvement in de novo generated offset responses. The potential role of other types of cortical neurons, like the somatostatin interneurons previously suggested to be involved in offset response generation (*Liu et al., 2019*), remains to be elucidated. It is also possible that information on sound termination arises from nonlemniscal areas projecting to AAF. Such possible projections and their potential contributions have not been described yet.

Second, the activity during WN stimulation in AAF shows a clear pattern of bursting activity. A transient onset is followed by a suppression phase, then rebound activity around 200 ms, followed by another suppression and rebound phase. Does this reflect an internal clock allowing AAF to follow sound duration irrespectively from inputs coming from MGB? What could be the role of such bursting activity in AAF and how could it be generated? Bursts are thought to be emitted by many subcortical and cortical areas of the brain, but their hypothesized functions differ across brain areas (*Zeldenrust et al., 2018*). It has previously been shown in marmosets that the ACx can use a combination of temporal and rate representation to encode a wide range of complex, time-varying sounds (*Lu et al.,*

*2001*). The offset responses and bursting activity we observe in the mouse AAF could play these multiple roles in encoding different temporal features such as ongoing sound and its termination. If the bursting and offset activity observed in AAF is a common feature of other primary sensory cortices is unclear.

Finally, we demonstrate that offset responses to WN or PT are strikingly different in the central auditory system (*Figure 7*). This could have multiple origins. First, the significantly increased activity of AAF neurons preceding WN termination could result in a decreased ability of neurons to respond properly to the end of the sound (*Figure 7j*). Second, the lack of proper inhibition – excitation balance (*Figure 7i*) could decrease the offset responses evoked by WN burst. WN stimulation is widely used in auditory research, especially for gap in noise detection (*Syka et al., 2002*; *Threlkeld et al., 2008*; *Weible et al., 2014a*; *Weible et al., 2014b*) and for offset response studies (*Anderson and Linden, 2016*). It is an attractive auditory stimulus to study purely temporal information as it ensures an effective stimulation of the auditory system irrespectively of neuronal tuning. However, one has to be careful about generalizing results obtained with this stimulus to all auditory inputs. Our observations indicate that different mechanisms might be at play when WN or PT, and by extension natural sounds, are heard.

The experiments with multifrequency component sounds show that within both MGB and AAF, offset responses indicate not only when a sound ends but also all important changes that occur within a temporally discontinuous sound (*Figure 8*), emphasizing their possible relevance for behavior and perception. The temporal integration of offset responses is crucial for the perceptual grouping of communication sounds, in which rapid changes in intensity and frequency occur (*Sollini et al., 2018*). Our results suggest that this integration is accentuated in the cortex, making it an interesting hub to look for the mechanisms that might explain impairments in sound offset sensitivity, and by extension, deficits in temporal processing arising both in aging and disease. A deeper knowledge of the cellular and circuit mechanisms of cortical offset responses could be crucial to develop new strategies to prevent abnormal auditory perceptual grouping.

# Materials and methods

**Key resources table**

| Reagent type (species) or resource | Designation | Source or reference | Identifiers | Additional information |
|---|---|---|---|---|
| Chemical compound, drug | Ketamin | Vetoquinol | ATCvet-Code: QN01A × 03 | |
| Chemical compound, drug | Xylazin hydrochlorine | Provet AG | ATCvet-Code: QN05CM92 | |
| Chemical compound, drug | Lidocain | Streuli | ATCvet-Code: QN01BB02 | |
| Chemical compound, drug | Bupivicaine | Sintetica | ATCvet-Code: N01BB01 | |
| Strain, strain background (*Mus musculus*) | Wild-type (C57BL/6J) | Janvier | C57BL/6JRj | |
| Strain, strain background (*M. musculus*) | PV-Cre | Jackson | Stock #017320 | *PValb*-Cre knock-in line |
| Strain, strain background (*M. musculus*) | Ai32 line | Jackson | Stock #024109 | ChR2-lox |
| Software, algorithm | Active X and RPvdsX data acquisition software | Tucker-Davis Technologies | | https://www.tdt.com |
| Software, algorithm | MATLAB | Mathworks | | https://www.mathworks.com/ |
| Software, algorithm | Kilosort | Github CortexLab | | https://github.com/cortex-lab/Kilosort (*Pachitariu et al., 2019*) |
| Software, algorithm | phy | Github CortexLab | | https://github.com/cortex-lab/phy (*Buccino et al., 2021*) |

*Continued on next page*

*Continued*

| Reagent type (species) or resource | Designation | Source or reference | Identifiers | Additional information |
|---|---|---|---|---|
| Software, algorithm | Logistic regression model | Github Neuromatch | | https://github.com/NeuromatchAcademy/course-content/blob/master/tutorials/README.md#w1d4---generalized-linear-models (*Fiquet et al., 2020*) |
| Other | Electrodes | Neuronexus | AA4 × 8-5 mm-50-200-177-A32 A1 × 32-5 mm-25-177-A32 A4 × 16-5 mm-50-200-177-A64 | |

## Surgical procedures

All experimental procedures were carried out in accordance with Basel University animal care and use guidelines and were approved by the Veterinary Office of the Canton Basel-Stadt, Switzerland (protocol cantonal number 2748). To target the opsins to PV+ interneurons, we used PV-Cre (Cre was targeted to the *Pvalb* locus) knock-in line with C57BL/6J background (JAX stock number 017320, Jackson Laboratories, ME, USA). This strain drives the expression of Cre in PV+ interneurons of the cortex with the minimal leak. We crossed this line to the Ai32 line (JAX stock number 024109 with C57BL/6 background), which encodes the light-gated depolarizing cation channel channelrhodopsin-2 conjugated to e-YFP after a floxed stop cassette under the CAG promoter. Wild types were C57BL/6J mice (Janvier, France). Thirty-nine mice were used in this study. Mice were a mixture of males and females and aged between 7 and 12 weeks of age at the time of behavioral training or electrophysiological recording.

Awake electrophysiology recordings and behavior experiments were performed on adult (7–12 weeks) male and female C57BL/6J mice (Janvier, France). For surgeries, mice were anesthetized with isoflurane (4% for induction, 1.5–2.5% for maintenance), and subcutaneous injection of bupivacaine/lidocaine (0.01 and 0.04 mg/animal, respectively) was used for analgesia. A custom-made metal head post was fixed with super glue (Henkel, Loctite) on the bone on top of the left hemisphere and used to head-fix the animals. Their body temperature was kept at 37°C with a heating pad (FHC, ME, USA), and lubricant ophthalmic ointment was applied on both eyes. Craniotomy (~2 × 2 mm$^2$) was performed with a scalpel just above the right ACx and covered with silicone oil and silicone casting compound (Kwik-Cast, World Precision Instruments, Inc, FL, USA) during the 2 hr recovery period from the anesthesia.

## Recordings

The electrophysiological recordings were performed in awake mice (AAF: $n$ = 6; MGB: $n$ = 5). Mice were head-fixed and placed in the cardboard tube for recordings inside a soundbox. Extracellular recordings were conducted in AAF (identified based on the functional tonotopy: ventrodorsal increase in BF) and MGB (centered 0.8 mm anterior and 2 mm lateral to Lambda). Multichannel extracellular electrodes with 32 channels (A4 × 8-5 mm-50-200-177-A32 or A1 × 32-5 mm-25-177-A32 Neuronexus, MI, USA) or 64 channels (A4 × 16-5 mm-50-200-177-A64, Neuronexus, MI, USA) were inserted orthogonally to the brain surface with a motorized stereotaxic micromanipulator (DMA-1511, Narishige, Japan) at a constant depth (AAF: the tip of the electrode at 556 ± 9 μm from pia; MGB: the tip of the electrode at 3575 ± 300 μm from pia). Responses from extracellular recordings were digitized with a 32- or 64-channel recording system (RZ5 Bioamp processor, Tucker Davis Technologies, FL, USA) at 24,414 Hz. Sorted units were identified from raw voltage traces using kilosort (*Pachitariu et al., 2016*; CortexLab, UCL, London, England) followed by manual corrections based on the interspike interval histogram and the consistency of the spike waveform (phy, CortexLab, UCL, London, England). All sorted data were used, independently

of whether the clusters were classified as single or multiunit. Further analysis was performed in MATLAB (Mathworks, MA, USA).

## Auditory stimulation

Sounds were generated with a digital signal processor (RZ6, Tucker Davis Technologies, FL, USA) at 200 kHz sampling rate and played through a calibrated MF1 speaker (Tucker Davis Technologies. FL, USA) positioned 10 cm from the mouse's left ear. Stimuli were calibrated with a wide-band ultrasonic acoustic sensor (Model 378C01, PCB Piezotronics, NY, USA).

## Antidromic stimulation

To identify temporal dynamics of cells projecting from MGB to AAF, in vivo electrophysiology recordings in MGB were combined with antidromic stimulation of AAF ($n$ = 5 animals). First, mice were anesthetized with an intraperitoneal injection of ketamine/xylazine (80 and 16 mg/kg, respectively), and subcutaneous injection of bupivacaine/lidocaine (0.01 and 0.04 mg/animal, respectively) was used for analgesia. Ketamine (45 mg/kg) was supplemented during surgery as needed. For surgery, mice were head-fixed, and their body temperature was kept at 37°C with a heating pad (FHC, ME, USA). Two separate craniotomies (~2 × 2 mm$^2$) were performed with a scalpel above the right MGB and ACx and covered with silicone oil. AAF was mapped with electrophysiology recordings based on the ventrodorsal increase in BF to identify the target area for stimulus pipette insertion. Then both craniotomies were covered with silicone casting compound (Kwik-Cast, World Precision Instruments, Inc, FL, USA) during the 2 hr recovery period from the anesthesia. Electric stimulator (Master-8, A.M.P.I., Israel) was connected to a stimulation isolator (ISO-Flex, A.M.P.I., Israel), which was then connected to the wire electrode. The wire electrode was fixed in pulled capillary glass (tip: <10 μm) filled with saline and then inserted into AAF (~300 μm). Monophasic square wave pulse was generated with electronic stimulator as pulse train (pulse duration: 0.1 ms; frequency: 50 Hz; train number: 20); intensity: 30 μA (similar to the method described in *Peng et al., 2017*). At the same time, electrophysiology recordings with 64-channel electrode were performed in MGB. As described in the recordings section, spike sorting was performed using kilosort (*Pachitariu et al., 2016*), followed by manual corrections in phy, and further analysis in MATLAB. To ensure the absence of electrical artifacts, the mean cluster waveform from raw data was calculated for each antidromic-identified cluster. Clusters containing any high amplitude electric artifacts were removed from the analysis.

## Offset detection task

*Headplate implant.* Mice were implanted with a custom-made metal head post at 7–8 weeks after birth under isoflurane anesthesia (4% induction, 1.2–2.5% maintenance). Local analgesia was provided with subcutaneous injection of bupivacaine/lidocaine (0.01 and 0.04 mg/animal, respectively). A head post and a ground screw were fixed to the skull with dental cement (Super-Bond C&B; Sun Medical, Shiga, Japan). The portion of the skull above the target recording site was left free from cement and covered with a thick layer of Kwik-Cast Sealant (WPI, Sarasota, FL, USA). Postoperative analgesia was provided with an intraperitoneal injection of buprenorphine (0.1 mg/kg). After recovery from the surgery for a couple of days, mice were food restricted. *Training.* Mice were then placed in the cardboard tube and adapted to the head restraint. The speaker was placed 10 cm away from the left ear of the animals. Next, they were taught to associate a sound offset with reward availability. Mice were trained to detect sound offset of PT (9 kHz) played at 80 dB SPL (training) with varied duration (1, 1.5, and 2 s). The rise-ramp of the tones was always fixed to 10.0 ms, while at the offset fast (0.01 ms) or slow (10.0 ms) ramp was used and varied randomly. During the beginning of the training, mice had to lick within 3 s after sound offset to receive a drop of soya milk as a reward, and the trial was considered a correct hit. If the animal did not lick within and after the tone trial was considered as missed. If the mice licked while the tone was ongoing, they received a mild air puff oriented toward the right eye and a time out (2–3 s) until the subsequent trial could start. These trials were removed from the analysis as the target (sound offset) could not be correctly delivered. Sounds were delivered without preceding cues at random ISIs ranging from 3 to 5 s. Licks were detected with a piezo sensor attached to the reward spout. Within consecutive training days, the reward window was shortened down to 1 s. *Craniotomy.* Once animals performed at least 30% of correct hits, they were considered initially trained and had a craniotomy performed under ketamine/xylazine (80 mg/kg) and AAF mapping on the same day. *Recordings.* On

the following day, mice were moved on to a tasting phase where behavior training was coupled with acute electrophysiology recordings in AAF. During the testing phase, tones were played at 60 dB SPL, and laser (473 nm) was added unilaterally above the right AAF and activated continuously for 200 ms following sound termination in half of the trials (pseudorandomized each day). All experiments were performed in a soundproof box (IAC acoustics, Hvidovre, Denmark) and monitored from outside the soundbox with a camera (C920, Logitech, Switzerland). The laser power was set around 4.2 mW and was adjusted every day to cause a robust suppression of offset or sustained responses in PV− cells. The testing phase was carried out for up to 6 days. Behavioral control and data collection were carried out with custom-written programs using a complex auditory processor (RZ6, Tucker Davis Technology, FL, USA) and further analyzed with MATLAB (MathWorks, MA, USA).

### Data analysis

All data analysis was performed using custom-written MATLAB (2019) (Mathworks) code. Original spike data and code are available on Dryad (https://doi.org/10.5061/dryad.41ns1rnfg).

### Tuning receptive fields

To determine BF and tuning receptive fields (TRFs), we used PT (50 ms duration, randomized ISI distributed equally between 500 and 1000 ms, two repetitions, 4 ms cosine on, and 0.01 ms cosine fall-ramps) varying in frequency from 4 to 48.5 kHz in 0.1-octave increments and in level from 0 to 80 dB SPL in 5 dB increments. TRFs, best frequency, and spiking rates were calculated in fixed time windows: onset: 6–56 ms, offset: 56–106 ms. TRFs were smoothed with a median filter (4 × 4 sampling window) and thresholded to 0.2 of peak amplitude. Onset and offset BF was defined as the frequency that elicited maximal response across all sound levels. Onset and offset peak latency was determined as the time point in which the smooth PSTH (kernel = hann (9)) collapsed across all tested stimuli showed a maximum response (binning size: 5 ms). Spontaneous activity was calculated based on activity preceding sound onset (150–50 ms, binning size: 5 ms).

### Tone duration responses

To study responses to tones with different durations, we used 10 repetitions of PT (AAF: 9 kHz; MGB: frequency adapted to offset BF of recorded neurons) with 4 ms cosine on and 0.01 ms cosine fall-ramps, which were varied in duration (50, 100, 150, 250, 350, and 500 ms), ISI (the gap between two stimuli of 50, 100, 250, 500, 1000, and 2000 ms) and played at 60 dB SPL. For AAF, the frequency was fixed to 9 kHz because 9 kHz PT evoked significant offset responses in almost all tested AAF. Offset spike rates were calculated in a fixed time window of 6–56 ms following sound termination.

### Spectral complexity

To study offset responses in MGB and AAF evoked by sounds with different spectral complexity, we recorded responses in both regions to 500 ms PT and white WN bursts played at 60 dB SPL with 4 ms cosine on and 0.01 ms cosine fall-ramps (for MGB the PT frequency was chosen based on offset BF of neurons in each session; for AAF, a fixed PT of 9 kHz was used). WN bursts were not fixed but consisted of randomly chosen noise samples. Offset spike rates were calculated in a fixed time window of 6–56 ms following sound termination.

### Sound rise- to fall-time study

To study the dependence of onset and offset responses on the temporal profile of a tone, we varied rise- and fall-time at sound onset and offset (0.01, 1, 2, 4, 10, 50, 100, and 200 ms). PTs (AAF: 9 kHz; MGB frequency adapted to offset BF of recorded neurons) were played at 60 dB SPL for 500 ms and repeated 50 times. The peak amplitude of offset responses was defined in the first 100 ms after stimulus onset or offset.

### Offset detection in ongoing sound

To check if offset responses encode changes within ongoing sound, we used tone consisting of three frequency components (20, 14, and 9 kHz) played at 60 dB SPL and repeated 60 times. All frequency components had common onset but were terminated at different time points (300, 400, and 500 ms).

Offset spike rates were calculated after every component removal in fixed windows: 306–356 ms; 406–456 ms; 506–556 ms.

## Sound termination detection task

Both the ODI and the hit rates were used to evaluate the animal's behavior in the sound termination detection task. ODI was defined as:

$$ODI = \frac{Hit + Correct\ Rejection}{Miss + False\ Alarm}$$

where *Hit* was calculated as the percent of 1-s long trials in which the mouse licked between 1 and 2 s (during the reward window), *Miss* as the percent of 1-s long trials in which the mouse did not lick between 1 and 2 s (during the reward window), *False Alarm* as the percent of 2-s long trials in which the mouse licked between 1 and 2 s (during the last 1 s of sound), and *Correct Rejection* as the percent of 2-s long trials in which the mouse did not lick between 1 and 2 s (during the last 1 s of sound). Trials with early onset licks (happening within 1 s following sound onset) and trials with 1.5 s long sounds were not taken into account for calculation of ODI. Similarly, trials with licks before sound termination were discarded from the hit rate analysis (*Figure 1—figure supplement 2c*). To compare ODI (*Figures 1 and 2*) or hit rate (*Figure 1—figure supplement 3*) for trials with and without laser or for different tested ramps, a moving average was calculated with a window size of 10 trials. The data for each condition were calculated separately. For the average, 10 adjacent trials were taken, and only the behavior corresponding to a specific condition were used (meaning that each average is made of a maximum of 10 trials but usually of less). This allows a comparison of performance over time across the tested conditions.

## Decoding population activity

The logistic regression model was used to decode animal performance from neural responses (code from Neuromatch Academy W1D4, https://academy.neuromatch.io/). Spontaneous activity (50–100 ms before sound onset), onset response (0–50 ms from sound onset), sustained response (500–450 ms before sound offset), offset response (0–50 ms from sound offset), or late response (500–550 ms after sound offset) were used to train and test the model. Logistic regression was implemented using the sklearn function LogisticRegression with the lbfgs solver and L2 regularization to avoid over-fitting. Eightfold cross validation was performed by leaving out a random 12.5% subset of trials to test the classifier performance, and remaining trials were used to train the classifier. A range of regularization values was tested (0.0001–10,000 log spaced), and the one that gave the smallest error on the validation dataset was chosen as the optimal regularization parameter. The classifier accuracy was computed as the percentage of testing trials in which the animal's choice was accurately predicted by the classifier and summarized as the average across the 10 repetitions of trial subsampling. The spiking activity of each neuron was *z*-scored before running the logistic regression model. Trial labels were shuffled to confirm that decoding is not working for random data. This procedure was repeated 10 times. Then the average across the 10 repetitions was used to assess the classifier accuracy for randomized data. To remove all the sessions with a too small number of trials or too few offset cells, only the sessions with a significant difference in classification accuracy between real and shuffled data based on the late response (0.5 s after sound offset) were used.

## Statistical analysis

Sample size was determined based on standards established by previous publications studying single-neuron activity with in vivo recording, which have been adequate to demonstrate significant population effects. A traditional power analysis is not possible because noise properties of neural data are difficult to estimate a priori. Based on norms for the field, we acquired data from at least five animals, except for *Figure 2b* (*n* = 2 animals), and numbering at least 14 behavioral sessions or 50 neurons per group, except for the number of putative FS neurons in *Figures 6 and 7* (*n* = 28). Statistical tests were performed with GraphPad Prism software version 7.03 (GraphPad Software, USA). The standard error of the mean was calculated to quantify the amount of variation between responses from different populations. PSTHs display (1) mean ± STD if they represent one condition, (2) mean ± standard error of mean if they represent more than one condition on

the same panel. A nonparametric, unpaired Mann–Whitney test was used to calculate whether there were any significant differences between medians of recordings in AAF and MGB. Wilcoxon paired test was used to compare differences between paired values obtained in different treatments. Friedman test with multiple comparisons was used to compare many conditions. Two-way ANOVA was used to test the main effects of sound duration and intervals on offset responses and their interaction effect. Dunn's multiple comparisons test was used to perform multiple pairwise comparisons. Spearman correlation tests were used to test for significant associations between pairs of variables measured with ranking. The effects were named significant if the p value was smaller than 0.05 (*), 0.01 (**), 0.001 (***), or 0.0001 (****), for a confidence interval of 95%, 99%, or 99.9%, respectively.

## Acknowledgements

This work was supported by grants from the Lundbeck Foundation (Fellowship to TRB) and the Swiss National Science Foundation (ERC Transfer grant to TRB). We thank Gioia De Franceschi for help with data analysis and Boris Gourévitch for thoughtful comments on the manuscript.

## Additional information

### Funding

| Funder | Grant reference number | Author |
|---|---|---|
| Lundbeckfonden | R139-2012-12375 | Tania Rinaldi Barkat |
| Schweizerischer Nationalfonds zur Förderung der Wissenschaftlichen Forschung | CRETP3-166735 | Tania Rinaldi Barkat |

The funders had no role in study design, data collection, and interpretation, or the decision to submit the work for publication.

### Author contributions

Magdalena Solyga, Conceptualization, Data curation, Formal analysis, Investigation, Writing - original draft; Tania Rinaldi Barkat, Conceptualization, Funding acquisition, Supervision, Writing - review and editing

### Author ORCIDs

Magdalena Solyga (iD) http://orcid.org/0000-0003-2969-2963
Tania Rinaldi Barkat (iD) http://orcid.org/0000-0001-8650-0986

### Ethics

All experimental procedures were carried out in accordance to Basel University animal care and use guidelines, and were approved by the Veterinary office of the Canton Basel-Stadt, Switzerland (protocol cantonal number 2748).

### Decision letter and Author response

Decision letter https://doi.org/10.7554/eLife.72240.sa1
Author response https://doi.org/10.7554/eLife.72240.sa2

## Additional files

### Supplementary files
• Transparent reporting form

### Data availability
Original spike data and code are available on Dryad (https://doi.org/10.5061/dryad.41ns1rnfg).

The following dataset was generated:

| Author(s) | Year | Dataset title | Dataset URL | Database and Identifier |
|-----------|------|---------------|-------------|-------------------------|
| Solyga M, Barkat T | 2021 | Emergence and function of cortical offset responses in sound termination detection | https://doi.org/10.5061/dryad.41ns1rnfg | Dryad Digital Repository, 10.5061/dryad.41ns1rnfg |

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
