## [Editor Report]

The work demonstrates specific neurophysiological cortical mechanisms for offset responses that are interesting in themselves. Two referees highlighted issues with the behavioural experiments that have been addressed in the revision. Reviewer #2 makes another minor suggestion that he authors might consider before publication of the final version.

---

## [Decision Letter]

**Decision letter after peer review:**

[Editors’ note: the authors submitted for reconsideration following the decision after peer review. What follows is the decision letter after the first round of review.]

Thank you for submitting your work entitled "Emergence and function of cortical offset responses in sound termination detection" for consideration by *eLife*. Your article has been reviewed by 3 peer reviewers, including Timothy D Griffiths as the Reviewing Editor and Reviewer #1, and the evaluation has been overseen by a Reviewing Editor and a Senior Editor.

We are sorry to say that, after consultation with the reviewers, we have decided that your work will not be considered further for publication by *eLife*.

The referees found the work very interesting. The neurophysiological data are excellent, but issues are raised with respect to the behavioral data. After discussion the referees all concur that these require further experiments and could not be addressed on the basis of the current data. For this reason, we have recommended rejection of the manuscript. We would be pleased to consider a resubmission that incorporated the additional data suggested.

*Reviewer #1:*

There has been a great deal of recent interest in the neural basis for offset responses given their hypothesised importance to perception. The possible behavioural relevance to cues like sound duration and gap duration has been taken as a self-evident truth in some work. I found this work attractive in actually testing the relevance of offset responses to duration perception in a mouse model in addition to examining the brain basis. The work is thorough and well executed. The work demonstrates offset responses that occurs for the first time in auditory cortex distinct from A1 where prevention of offsets by activating cells causes worsening of behavioural performance.

1. I think an initial concern in discussion of this manuscript about artefactual effects of spectral splatter due to abrupt sound termination have been addressed in this version of the manuscript.

2. The data appear to support a specialisation for offset response in AAF but have offset responses and their behavioural relevance been examined in A1? The analysis in figure 4 convincingly demonstrates changes in offset responses between MGB and AAF (which is monosynaptic connection) but it would be interesting to know about A1. I appreciate there is s strong prior related to AAF based on previous work but the offset responses being in AAF has been almost taken as a given in manuscript.

*Reviewer #2:*

In this study, the authors examine: (1) whether offset responses, where neurons respond upon termination of a stimulus, are behaviorally relevant; (2) whether offset responses are merely inherited from subcortical stations or are generated and amplified in cortex; and (3) whether offset responses simply encode sound termination or if they carry stimulus identity information as well. They show, using a combination of optogenetics and behavior, that suppressing offset responses in auditory field AAF results in an impairment of sound termination detection. They then show, using single- and multi-unit recordings, that the behavioral choice of the animal can be decoded on a trial-by-trial basis from the offset and late response phases. Finally, using antidromic stimulation and using multiple stimuli, the authors show that AAF offset responses are not wholly inherited from the auditory thalamus.

The electrophysiological elements of the study seem solid and well-performed. Some weaknesses of the study include the effectiveness of task acquisition by the behavioral subjects, and behavioral analyses that discard trials with potentially useful information. Some statistical tests may not be appropriate and brings into question the results of the decoding analysis. Very recent and highly relevant publications are not discussed in the study. Additional control analyses would strengthen the manuscript.

1. Two very recent studies address questions that are central to this manuscript. First, Li H et al. (Cell Reports, 2021, 35:109003) show using optogenetic manipulations in primary auditory cortex (A1) that A1 OFF activity is required for the perception of sound duration. These results must be discussed in the context of the authors claim that AAF might be specialized for the detection of offset responses.

2. Second, Bondanelli et al. (*Elife*, 2021, 10:e53151) argue for a role of recurrent A1 connectivity in shaping offset responses in cortex, including the fact that the offset response carries information about stimulus type. These results should be discussed in the context of the authors observations as well.

3. Regarding behavior: the authors discard trials from analyses when the animal licked while the tone was ongoing, and this appears problematic. From the description in the methods, it is unknown what fraction of total trials were discarded from analyses. These trials could be coded as false alarms, and when this information is included in the analysis by using a metric such as the sensitivity index (d’), could provide a complete picture of the behavior.

Considering 30% correct trials as ‘trained’ seems well below traditionally accepted metrics of when a animal is considered trained, especially for a relatively simple detection task. Usually, this number is closer to 70% correct – for example, in the Li et al. 2021 paper mentioned above, mice were considered trained after reaching 90% correct trials on a sound duration discrimination task. Better yet, a d-prime of 1 or 1.5 when false alarms are also considered is a more sensitive metric of behavior (for example, see Caras and Sanes, J. Neurosci 2015).

4. In decoding of performance from activity, given that the reward window opens at offset and is open for only 1 s, the inclusion of the ‘late’ phase is problematic unless it can be shown that licks do not occur within 0.5 s of sound offset. This bump for the hits could result from multiple effects – movement, reward, licking sounds etc. The data supporting the claim of better decoding from offset responses hinges on Figure 3c, where offset responses yield greater accuracy than onset responses. However, pairwise Wilcoxon tests do not seem appropriate for these data where multiple comparisons are being made. The authors should use an ANOVA or Kruskal-Wallis test followed by by multiple-comparisons corrected posthoc tests.

5. From Figure 1E, it appears that the post-inhibitory rebound in other cells in the laser on condition has a similar magnitude to the offset response in the laser off condition. Could the rebound be driving AAF responses that signal an offset, albeit delayed by about 0.2 s, that the animals could be using to detect sound termination? To answer this question, could the authors analyze both the neurophysiological data, as well as determine if the correct responses in the laser ON condition have longer latencies consistent with this 0.2 s delay?

6. If the authors have the data available, it would be great to see a similar control as shown in Figure 2j-m for the longer ramp duration as well in Figure 1. More detail in the methods section as to how the fiber was placed over AAF (in craniotomy but above dura?), whether it was optically shielded to prevent visual cues etc. would be helpful.

7. For the onset-offset neurons that do not have a sustained response profile, it is clear that the highly correlated offset is an important distinguishing cue – it provides a high-SNR signal between the offset response and the previous silent period (when the tone is on). But what if (as for the white noise stimuli, Figure 7b) some amount of sustained activity is present? Is offset-detection behavior worse, and is decoding accuracy using the classifier also worse? If behavioral data is not available, could additional analyses be performed to predict sound termination time for pure tones and white noise, and make a prediction as to what would happen behaviorally?

*Reviewer #3:*

The goal of this study was to assess the function of cortical offset responses of the anterior auditory field (AAF) in sound perception. The authors used a combination of behavioral, electrophysiological and optogenetic techniques to study the properties of cortical offset responses. Through behavioral experiments combined with optogenetics, the authors first claim to find that inhibiting offset responses in the AAF decrease the mouse’s ability to detect when a sound ends. Furthermore, they report that larger offset responses correlate with an increase in the mouse’s ability to detect sound termination. Functionally, the authors demonstrate via electrophysiological experiments that cortical offset responses have a component that is generated in the AAF and therefore not only inherited from the periphery. The authors also find that offset responses increase with sounds that have longer duration and therefore do not simply encode for silence. The electrophysiological investigation of the properties of cortical offset responses is well designed and the conclusions are justified by the data. However, several questions about the behavioral paradigm arose that warrant further control experiments and re-examining interpretation.

1) The behavioral paradigm suffers from a design in which it is difficult to estimate the false alarm rate. Therefore, it is unclear whether the mouse is trained to lick in response to tone offsets, or rather to reduce licking during the sound presentation. The criterion for “fully trained” is set at 30% hit rate, well below chance (Figure 1b), which seems somewhat low. It is unclear what the mouse licking is reporting.

2) This interpretation of behavioral performance is complicated by the high rate at which mice licked during the trials (Figure S1). (Is the legend for figure S1 correct?) Since the authors report that “Trials with licks at the tone onset were discarded from the analysis (Figure S1)”, question arises whether 60% of the trials were excluded for some mice. In that case, is the lick rate on the trials that were kept simply by chance?

3) The optogenetic manipulation of AAF during behavior drives a relatively small (20%) reduction in performance (Figure 1h and Figure 2i). It is therefore unclear that the conclusion that AAF is necessary for sound offset detection is supported by the data. The behavioral + optogenetic paradigm should be better described in methods, allowing to better assess the presentation of the laser (see point 8).

4) The reduction in performance may be due to general laser presentation and not specific to sound offset. It would be useful to present the laser during the sound or at different points in the stimulus to better understand how its presentation relates to offset responses specifically (Figure 1 and Figure 2).

5) It would be helpful to present more details about how the classifier was trained and tested (Figure 3). The neuronal classifier also does not seem to predict accurately behavior as the accuracy for the offset responses is at ~65 % (Figure 3c). It is furthermore interesting that much of behavior can be explained based on spontaneous neuronal activity, and therefore an alternative interpretation would be that cortical state (or another factor that accounts for differences in spontaneous activity between trials). This also suggests that a control as suggested in point 4 (laser presented at different time points in the stimulus) would be useful, as the classifier would predict significant reduction in performance based on suppression of activity prior to sound, and not at sound offset.

[Editors’ note: further revisions were suggested prior to acceptance, as described below.]

Thank you for resubmitting your work entitled “Emergence and function of cortical offset responses in sound termination detection” for further consideration by *eLife*. Your revised article has been evaluated by Barbara Shinn-Cunningham (Senior Editor) and a Reviewing Editor.

The manuscript has been improved but there are some remaining issues that need to be addressed, as outlined below. *eLife* makes great efforts to avoid multiple cycles of revision but referee #2 and referee #3 raise further important points in their reviews about the behavioural data that need to be addressed.

*Reviewer #2:*

I appreciate the authors’ detailed responses to the concerns arising from the earlier review of this manuscript. There are no major concerns with the electrophysiology part of the revised manuscript.

Behavior: My concerns with behavioral experiments and what the mice are actually reporting in the task remain.

1) I respectfully disagree with the authors’ statement that false alarm rates cannot be calculated for detection tasks. The term ‘False Alarm’ is derived from signal detection theory, where the subject reports that a signal was present when a signal was not actually present. In this case, the signal is the offset. If a mouse reports an offset by licking during the sound (when an offset, i.e. the signal, was not present), that outcome should be a false alarm. In the authors’ task design, only correct rejections cannot be specified (to do so, one would have to define a trial duration independent of sound termination time. If the mouse withholds licking for the trial duration while the sound has not actually terminated, it would be a correct rejection). I agree with the authors that if the mouse starts licking before even the sound onset, those trials should be discarded.

2) “Likewise, unlike in a discrimination task, a hit rate of 50% is not a chance level in a detection task.”

– It is not a question of what the chance level was, but rather a question of whether the mouse was reliably detecting the offset. P(Hit) + p(FA) + p(Miss) + p(CR) = 1, and a measure of performance would take into account the ratio of desirable (p(Hit)+p(CR)) and undesirable (p(FA)+p(Miss)) outcomes. To illustrate, in Figure S2 panel h, I count 50 trials, with n(Hit) = 23, n(Miss) = 11, nFA = 16, and n(CR) = 0. In this counting, the number of desirable outcomes (23+0=23) is lesser than the number of undesirable outcomes (11+16 = 27) and would result in a d’ = z(Hits) – z(FA) = 0.37. However, if trials in which animals licked during the sound (which I coded as FA) are dropped, then n(Total) = 50-16 = 34, with n(Hits) = 23 (23/34 = 67%). This is why dropping the onset lick trials and considering 30% hits as reaching criterion for training is not convincing.

Other metrics to better capture the behavior might be a discrimination index as used in, for example, Schwartz and David, 2018. Here the cumulative lick probability can be used to calculate ROC curves.

3) Page 5, lines 121 and 125: The authors use a very similar effect size change in hit rate for PV-Cre animals (4.5 +/- 2%) and for wild type animals (4.4 +/- 2.4%) to make the claim that there was a significant change in hit rate (p = 0.0264) for PV-cre animals, but no change (p=0.058) for the wild type animals (Figures 1H and 1L). The appropriate comparison in this case would be between 1H and 1L. (similar analysis must be performed comparing Figure 2I and 2M). It would be instructive if the authors could plot the Laser OFF and ON hit rates separately for the two groups (PV-Cre and wild type animals) so that one can also appreciate if there are overall differences in performance levels.

Additionally, the sizes of the error bars in Figure 1H and 1L are inconsistent with the numbers reported (why does 1H have the larger error bars despite the smaller value of 2% reported in main text)?

Citing recent literature:

I appreciate the authors’ inclusion of two recent studies in the Discussion section. However, these studies also affect the framing of the manuscript in the introduction section.

Page 2, line 39: “De-novo generation or amplification of offset responses in these areas have not been demonstrated yet.” – Please rephrase and appropriately cite the Bondanelli 2021 study here.

Page 2, lines 42 – 52: In this paragraph discussing the perceptual significance of offset responses, please appropriately discuss the Li 2021 study. Please rephrase the last sentence of this paragraph.

Page 4, line 86: Please rephrase “Changing the neuronal activity of sound offset responses without changing any other parameters of the sound response has not been tested” in light of the Li 2021 study.

Decoding analysis:

The classifier accuracy is not significantly different between the offset and onset windows, which suggests that onset responses can be as predictive of the mouse’s offset detection behavior as offset responses. Could the authors please discuss?

*Reviewer #3:*

The paper represents an innovative and comprehensive body of work aimed to assess the function of cortical offset responses in the anterior auditory field in sound perception. Whereas the majority of work to date in auditory neuroscience has focused on the sound onset responses (largely due to the quick adaptation of cortical responses), the offset responses, which are present in the cortex, and especially, as the authors show, in AAF, have received less attention. The offset responses can play an important role in sound segregation and auditory scene analysis. The authors used a combination of behavioral, electrophysiological and optogenetic techniques to study the properties of cortical offset responses. The authors first test whether and how offset responses correlate and affect behavioral detection of sound offsets. They find that suppressing offset responses in AAF reduces the responses of mice to sound offsets and that there is a significant correlation between cortical responses and behavioral report. The authors next use elegant electrophysiological and manipulation methods to find that cortical offset responses have a component that is generated in the AAF and therefore not only inherited from the periphery. The authors also find that offset responses increase with sounds that have longer duration and therefore do not simply encode for silence. Behaviorally, authors provide evidence for a role of cortical offset responses in sound termination perception. Such an extensive description of these types of responses provides for a substantial advance in our understanding of cortical function.

The authors have conducted additional experiments and analysis and the revised manuscript is much improved. We have only 1 outstanding concern.

Responses to 1+2: We believe that it is important to consider not only the hit rate, but also other signal detection measures in interpreting mouse behavior. In fact, trials on which the mouse licked during sound may be informative, and I am not sure it’s warranted to not include information about them in the analysis. One possible approach would be to compute signal detection theory measures of HR (hit rate), CR (correct rejects), M (misses rate) and FA (false alarm rate), from which one can compute d’ using standard approaches.

– Consider only the trials in which the sound is presented for either 1 s or 2 s.

– To compute hits, compute the percent of 1s-long trials, in which the mouse licked between 1s and 2 s (during the reward window).

– To compute misses, compute the percent of 1s-long trials, in which the mouse did not lick between 1s and 2 s (during the reward window).

– To compute FA, compute the percent of 2s-long trials, in which the mouse licked between 1s and 2s (during the last 1s of sound).

– To compute CR, compute the percent of 2s-long trials, in which the mouse did not lick between 1s and 2s (during the last 1s of sound).

Response to 10. Please use <= 2 significant digits for p values (e.g. p = 0.5655 should be reported as p=0.57 on line 415).

---

## [Author Response]

[Editors’ note: the authors resubmitted a revised version of the paper for consideration. What follows is the authors’ response to the first round of review.]

Reviewer #1:There has been a great deal of recent interest in the neural basis for offset responses given their hypothesised importance to perception. The possible behavioural relevance to cues like sound duration and gap duration has been taken as a self-evident truth in some work. I found this work attractive in actually testing the relevance of offset responses to duration perception in a mouse model in addition to examining the brain basis. The work is thorough and well executed. The work demonstrates offset responses that occurs for the first time in auditory cortex distinct from A1 where prevention of offsets by activating cells causes worsening of behavioural performance.1. I think an initial concern in discussion of this manuscript about artefactual effects of spectral splatter due to abrupt sound termination have been addressed in this version of the manuscript.

We are pleased to read that the description of the possible effects of spectral splatter, as described in the submitted manuscript, removed this concern.

2. The data appear to support a specialisation for offset response in AAF but have offset responses and their behavioural relevance been examined in A1? The analysis in figure 4 convincingly demonstrates changes in offset responses between MGB and AAF (which is monosynaptic connection) but it would be interesting to know about A1. I appreciate there is a strong prior related to AAF based on previous work but the offset responses being in AAF has been almost taken as a given in manuscript.

We thank the reviewer for this comment. We analyzed offset responses from awake recordings in both A1 and AAF for sounds with a duration of 500 ms (Figure 1—figure supplement 1). As offset responses at the population level are minimal in A1 compared to AAF, we think that their contribution in a sound detection task will be much more difficult to evaluate. We, therefore, decided to focus on AAF in this manuscript.

In the revised manuscript, this information has now been added (l.91-95), as Figure S1.

(l.91-95) “As, at the cortical level, transient offset responses cluster within AAF rather than A1 (Solyga and Barkat, 2019) and are absent in the averaged population activity of A1 neurons in response to 9 kHz pure tones in awake preparations (Figure S1), we decided to focus on AAF to assess the behavioral relevance of cortical offset responses.”

Reviewer #2:In this study, the authors examine: (1) whether offset responses, where neurons respond upon termination of a stimulus, are behaviorally relevant; (2) whether offset responses are merely inherited from subcortical stations or are generated and amplified in cortex; and (3) whether offset responses simply encode sound termination or if they carry stimulus identity information as well. They show, using a combination of optogenetics and behavior, that suppressing offset responses in auditory field AAF results in an impairment of sound termination detection. They then show, using single- and multi-unit recordings, that the behavioral choice of the animal can be decoded on a trial-by-trial basis from the offset and late response phases. Finally, using antidromic stimulation and using multiple stimuli, the authors show that AAF offset responses are not wholly inherited from the auditory thalamus.The electrophysiological elements of the study seem solid and well-performed. Some weaknesses of the study include the effectiveness of task acquisition by the behavioral subjects, and behavioral analyses that discard trials with potentially useful information. Some statistical tests may not be appropriate and brings into question the results of the decoding analysis. Very recent and highly relevant publications are not discussed in the study. Additional control analyses would strengthen the manuscript.1. Two very recent studies address questions that are central to this manuscript. First, Li H et al. (Cell Reports, 2021, 35:109003) show using optogenetic manipulations in primary auditory cortex (A1) that A1 OFF activity is required for the perception of sound duration. These results must be discussed in the context of the authors claim that AAF might be specialized for the detection of offset responses.

These two papers were not published yet when we initially submitted our manuscript. We fully agree that they are relevant to this manuscript and should be cited and discussed.

In the revised manuscript, the Li et al. paper is discussed on l.565-570 and 610-613:

(l.565-570) “It was recently shown that A1 transient offset responses to noise stimulation underlie the perception of sound duration (Li et al., 2021). As AAF supplies <inline-graphic mime-subtype="png" mimetype="image" xlink:href="media/image1.png" />45% of the cortical input to A1 (Lee and Winer, 2008) and the modulation of acoustic information between A1 and AAF in the cat’s Acx was shown to be dominated by a unidirectional AAF to A1 pathway (Carrasco and Lomber, 2009), it would be interesting to understand if A1 offset responses originate from AAF.”

(l. 610-613) “Additionally, the lack of increase in offset response amplitude with the duration of WN bursts (Figure S10) suggests that a stimulus has to contain a spectral structure to evoke duration dependent offset responses (Li et al., 2021).”

2. Second, Bondanelli et al. (Elife, 2021, 10:e53151) argue for a role of recurrent A1 connectivity in shaping offset responses in cortex, including the fact that the offset response carries information about stimulus type. These results should be discussed in the context of the authors observations as well.

In the revised manuscript, the Bondanelli et al. paper is discussed on l. 570-573:

(l. 570-573) “Very recently, Bondanelli et al. argued for a role of recurrent A1 connectivity in shaping offset responses in cortex, including the fact that the offset response carries information about stimulus type (Bondanelli et al., 2021). The high level of information exchange between A1 and AAF raises the question of whether these mechanisms in A1 depend on AAF activity.”

3. Regarding behavior: the authors discard trials from analyses when the animal licked while the tone was ongoing, and this appears problematic. From the description in the methods, it is unknown what fraction of total trials were discarded from analyses. These trials could be coded as false alarms, and when this information is included in the analysis by using a metric such as the sensitivity index (d’), could provide a complete picture of the behavior.Considering 30% correct trials as ‘trained’ seems well below traditionally accepted metrics of when a animal is considered trained, especially for a relatively simple detection task. Usually, this number is closer to 70% correct – for example, in the Li et al. 2021 paper mentioned above, mice were considered trained after reaching 90% correct trials on a sound duration discrimination task. Better yet, a d-prime of 1 or 1.5 when false alarms are also considered is a more sensitive metric of behavior (for example, see Caras and Sanes, J. Neurosci 2015).

We thank the reviewer for this comment. We fully agree that the analysis of the behavioral experiments would benefit from the calculation of a false alarm rate and that the original manuscript was lacking a detailed description of the behavioral paradigm. However, false alarm rates can be calculated for discrimination tasks (Li et al., 2021 or Caras and Sanes, 2015) but not for detection tasks. Likewise, unlike in a discrimination task, a hit rate of 50% is not a chance level in a detection task. To assess the rate at which mice would lick in case they would not detect sound termination, we calculated the lick rate in a window following the reward window and defined this as the chance level or random lick rate. Comparing this random lick rate with the hit rate (licks during the reward window) allows us to demonstrate that mice were performing the task and could detect sound termination, well above chance (Figure 1—figure supplement 2). This confirms the high relevance of our behavioral data.

To quantify the behavioral influence of offset responses, especially when manipulating them such as what is done with optogenetics in Figure 1 or with offset ramp duration in Figure 2, the mice had to be able to hear sound termination and to be ready to react. If they licked before sound termination, that would not be possible. We, therefore, removed all trials where the mice licked before sound termination. Compared to a normal detection task (detection at the onset of a sound), that would be analogous to remove the trials where the animals would lick before the stimulus, as is commonly done in the field.

In the revised manuscript, we have:

– Added Figure S2 and updated Figure 1.

– Added the corresponding information on .103-107.

(l.103-107) “The chance level of licks was calculated in a late window following the end of the reward window (Figure S2b, d, h). An increase in correct offset detection (hit rate), a decrease in reaction time over the training sessions (Figure 1d), and a significant difference between hit rate and licks at the chance level (Figure S2e-g) reflected successful learning after which animals were moved to the test phase (Figure 1b).”

4. In decoding of performance from activity, given that the reward window opens at offset and is open for only 1 s, the inclusion of the ‘late’ phase is problematic unless it can be shown that licks do not occur within 0.5 s of sound offset. This bump for the hits could result from multiple effects – movement, reward, licking sounds etc. The data supporting the claim of better decoding from offset responses hinges on Figure 3c, where offset responses yield greater accuracy than onset responses. However, pairwise Wilcoxon tests do not seem appropriate for these data where multiple comparisons are being made. The authors should use an ANOVA or Kruskal-Wallis test followed by by multiple-comparisons corrected posthoc tests.

We agree that the ‘late’ bump in the hit trials is probably due to movement or reward, and discuss this on l. 560-562:

(l.560-562) “The neural activity during hit trials in AAF could be influenced by the animal’s general motivation (Fritz et al., 2003), motor-related inputs (Schneider, 2020), or reward expectations (De Franceschi and Barkat, 2020).”

The decoding performed on the ‘late’ phase serves as a control, confirming that specific patterns of activity within individual neurons reflect an action of the animal, hence leading to better decoding.

Regarding statistics, we thank the reviewer for noticing the inappropriateness of the test previously used. We have now used the Friedman test (nonparametric ANOVA) with multiple comparisons. Figure 3 has been updated accordingly in the revised manuscript.

5. From Figure 1E, it appears that the post-inhibitory rebound in other cells in the laser on condition has a similar magnitude to the offset response in the laser off condition. Could the rebound be driving AAF responses that signal an offset, albeit delayed by about 0.2 s, that the animals could be using to detect sound termination? To answer this question, could the authors analyze both the neurophysiological data, as well as determine if the correct responses in the laser ON condition have longer latencies consistent with this 0.2 s delay?

We have now compared the reaction times for trials with and without laser. We did not see any difference in reaction times for trials with and without laser when sounds were terminated with a fast ramp. Interestingly sounds terminated with a slower ramp with laser stimulation were detected significantly slower than when the laser was not applied (Figure 1—figure supplement 4).

This delay in reaction time due to the laser is 15 ms on average. If this delay were due to the reaction to the rebound created by the laser, we would expect the delay to be of the duration of the laser application, i.e., 200 ms. The rebound can, therefore, not justify this reaction time difference. We think that the delay could instead be explained by the fact that with the laser on, offset responses are decreased and the sound termination detection therefore more difficult, leading to a delayed behavioral response.

In the revised manuscript, Figure 1—figure supplement 4 has been added as Figure S3. The corresponding description is on l.121 and l.214:

(l.121-122) “Interestingly, reaction times were significantly longer in laser on compared to laser off trials (Figure S3).”

(l.214-215) “There was no significant difference in the reaction times for both tested ramps (Figure S3).”

6. If the authors have the data available, it would be great to see a similar control as shown in Figure 2j-m for the longer ramp duration as well in Figure 1. More detail in the methods section as to how the fiber was placed over AAF (in craniotomy but above dura?), whether it was optically shielded to prevent visual cues etc. would be helpful.

We have now added the light-control experiment results also for the long ramp (10 ms). In the revised manuscript, they are now integrated within Figure 1.

All our behavioral experiments with optogenetic manipulations were coupled with electrophysiological recording. For all behavioral sessions, we first identified AAF based on the tonotopic gradient obtained with electrophysiological recordings. We then placed an optical fiber above AAF (above the dura) and titrated the laser (473 nm) power such that it would remove offset responses. Finally, we performed the behavioral session with these specific optogenetic parameters (optic fiber placement and laser power) and confirmed at the end of each session that offset responses were removed during the laseron trials. No optical shield was used. Control experiments confirmed that the laser did not give any visual cue (Figure 1i-l, Figure 2j-m).

In the revised manuscript, we have added this information in the Results section:

(l.109-116) “All our behavioral experiments with optogenetic manipulations were coupled with electrophysiological recording. For all behavioral sessions, we first identified AAF based on the tonotopic gradient obtained with electrophysiological recordings. We then placed an optical fiber above AAF (above the dura) and titrated the laser (473 nm) power to remove offset responses. Finally, we performed the behavioral session with these specific optogenetic parameters (optic fiber placement and laser power). We confirmed at the end of each session that offset responses were removed during the laser-on trials. No optical shield was used.”

7. For the onset-offset neurons that do not have a sustained response profile, it is clear that the highly correlated offset is an important distinguishing cue – it provides a high-SNR signal between the offset response and the previous silent period (when the tone is on). But what if (as for the white noise stimuli, Figure 7b) some amount of sustained activity is present? Is offset-detection behavior worse, and is decoding accuracy using the classifier also worse? If behavioral data is not available, could additional analyses be performed to predict sound termination time for pure tones and white noise, and make a prediction as to what would happen behaviorally?

As offset responses are much smaller for white noise than for pure tones, we believe that their contribution in the sound detection task will be much smaller than the larger offset responses for pure tones. Additionally, offset responses evoked by WN bursts are not increasing as a function of sound duration (Figure 7—figure supplement 2). On the other hand, there is a strong ongoing activity in AAF when longer sounds are presented (Figure 7—figure supplement 2d), raising the question of offset response relevance for this type of sound. We agree that evaluating the role of this sustained activity in the sound detection task is a most interesting point, but we feel that it goes beyond the scope of this manuscript, which is to evaluate the role of offset responses in particular. To understand to which extent onset, sustained, and offset activity contribute to sound termination perception and whether the contribution of all response types differs as a function of the spectral complexity of the stimuli requires significantly more behavioral studies. We are however following up on the role of sustained activity and its relation to onset and offset responses, but plan to address this in another study.

We refer to this figure in the results and Discussion sections.

(l.471-473) “This extended firing could affect the generation of offset for WN stimulation. Offset responses evoked by WN were not increasing as a function of sound duration in neither MGB nor AAF (Figure S10).”

(l.610-613) "Additionally, the lack of increase in offset response amplitude with the duration of WN bursts (Figure S10) suggests that a stimulus has to contain a spectral structure to evoke duration dependent offset responses (Li et al., 2021). "

Reviewer #3:The goal of this study was to assess the function of cortical offset responses of the anterior auditory field (AAF) in sound perception. The authors used a combination of behavioral, electrophysiological and optogenetic techniques to study the properties of cortical offset responses. Through behavioral experiments combined with optogenetics, the authors first claim to find that inhibiting offset responses in the AAF decrease the mouse's ability to detect when a sound ends. Furthermore, they report that larger offset responses correlate with an increase in the mouse's ability to detect sound termination. Functionally, the authors demonstrate via electrophysiological experiments that cortical offset responses have a component that is generated in the AAF and therefore not only inherited from the periphery. The authors also find that offset responses increase with sounds that have longer duration and therefore do not simply encode for silence. The electrophysiological investigation of the properties of cortical offset responses is well designed and the conclusions are justified by the data. However, several questions about the behavioral paradigm arose that warrant further control experiments and re-examining interpretation.1) The behavioral paradigm suffers from a design in which it is difficult to estimate the false alarm rate. Therefore, it is unclear whether the mouse is trained to lick in response to tone offsets, or rather to reduce licking during the sound presentation. The criterion for "fully trained" is set at 30% hit rate, well below chance (Figure 1b), which seems somewhat low. It is unclear what the mouse licking is reporting.

We thank the reviewer for this comment. We fully agree that the analysis of the behavioral experiments would benefit from the calculation of a false alarm rate and that the original manuscript was lacking a detailed description of the behavioral paradigm. However, false alarm rates can be calculated for discrimination tasks but not for detection tasks. Likewise, unlike in a discrimination task, a hit rate of 50% is not a chance level in a detection task. To assess the rate at which mice would lick in case they would not be able to detect sound termination, we calculated the lick rate in a window following the reward window and defined this as the chance level or random lick rate. Comparing this random lick rate with the hit rate (licks during the reward window) allows us to clearly demonstrate that mice were really performing the task and were able to detect sound termination, well above chance (Figure 1—figure supplement 1d-n). This confirms the high relevance of our behavioral data.

In the revised manuscript, we have:

– Added Figure S2 and updated Figure 1.

– Added the corresponding information on l.103-107.

(l.103-107) "The chance level of licks was calculated in a late window following the end of the reward window (Figure S2b, d, h). An increase in correct offset detection (hit rate), a decrease in reaction time over the training sessions (Figure 1d), and a significant difference between hit rate and licks at the chance level (Figure S2e-g) reflected successful learning after which animals were moved to the test phase (Figure 1b)."

2) This interpretation of behavioral performance is complicated by the high rate at which mice licked during the trials (Figure S1). (Is the legend for figure S1 correct?) Since the authors report that "Trials with licks at the tone onset were discarded from the analysis (Figure S1)", question arises whether 60% of the trials were excluded for some mice. In that case, is the lick rate on the trials that were kept simply by chance?

We explain in the previous point how we now calculate the chance level.

To quantify the behavioral influence of offset responses, especially when manipulating them such as what is done with optogenetics in Figure 1 or with offset ramp duration in Figure 2, the mice had to be able to hear sound termination and to be ready to react. If they licked before sound termination, that would not be possible. We, therefore, discarded all trials where the mice licked before sound termination. Compared to a normal detection task (detection at the onset of a sound), that would be analogous to remove the trials where the animals would lick before the stimulus, as is commonly done in the field.

3) The optogenetic manipulation of AAF during behavior drives a relatively small (20%) reduction in performance (Figure 1h and Figure 2i). It is therefore unclear that the conclusion that AAF is necessary for sound offset detection is supported by the data. The behavioral + optogenetic paradigm should be better described in methods, allowing to better assess the presentation of the laser (see point 8).

We agree with the reviewer that it is not possible to conclude that AAF is necessary for sound offset detection. What we claim is that AAF offset responses help sound termination detection and that they have a behavioral role, as clearly stated on l.15, l.120, l.258, or l.541 of the manuscript – not that they are necessary. As discussed, information from non-lemniscal areas (l.629-631) or sustained activity (l.604-606) could, for example, also play a role.

The relatively small reduction in performance (20 %) should be interpreted knowing that the laser light was applied unilaterally – this number might increase if the manipulation is bilateral. Despite this, our data clearly demonstrate that transient offset responses in AAF improve the detection of sound termination.

All our behavioral experiments with optogenetic manipulations were coupled with electrophysiological recording. For all behavioral sessions, we first identified AAF based on the tonotopic gradient obtained with electrophysiological recordings. We then placed an optical fiber above AAF (above the dura) and titrated the laser (473 nm) power such that it would remove offset responses. Finally, we performed the behavioral session with these specific optogenetic parameters (optic fiber placement and laser power) and confirmed at the end of each session that offset responses were removed during the laseron trials. No optical shield was used. Control experiments confirmed that the laser did not give any visual cue (Figure 1i-l, Figure 2j-m).

In the revised manuscript, we have added this information in the Results section:

(l.109-116) "All our behavioral experiments with optogenetic manipulations were coupled with electrophysiological recording. For all behavioral sessions, we first identified AAF based on the tonotopic gradient obtained with electrophysiological recordings. We then placed an optical fiber above AAF (above the dura) and titrated the laser (473 nm) power to remove offset responses. Finally, we performed the behavioral session with these specific optogenetic parameters (optic fiber placement and laser power). We confirmed at the end of each session that offset responses were removed during the laser-on trials. No optical shield was used."

4) The reduction in performance may be due to general laser presentation and not specific to sound offset. It would be useful to present the laser during the sound or at different points in the stimulus to better understand how its presentation relates to offset responses specifically (Figure 1 and Figure 2).

First, we hope that the more detailed explanation of our behavioral experiment (Figure 1—figure supplement 2) will clarify how the optogenetic manipulations were performed and how the results relate to offset responses specifically. Then, we have performed control experiments to show that the laser itself did not cause any change in behavioral performance and that it is really its effect on offset responses that explains the change in behavioral performance (Figure 1, Figure 1m). Finally, we have performed additional experiments in PV-ChR2 animals where the laser light was applied for 200 ms during the sound presentation (Figure 1m and Figure 2). The laser light (473 nm) was applied for 200 ms starting at 200 ms following sound onset. The laser light activated PV+ cells in AAF, which in turn suppressed the activity of other AAF cells (Figure 1n, Figure 2o). This laser manipulation did not result in any specific change in the animals’ performance (Figure R7c-d, g-h), strengthening our results with the laser application at the sound termination.

These data are now incorporated within Figures 1 and 2 and described in the result section l.126-132 and l.262-268.

(l.126-132) "Finally, we performed experiments in PV-ChR2 animals where the laser light was applied for 200 ms during the sound presentation and evaluated the animals' performance in the sound termination detection task (Figure 1m). The laser light was applied for 200 ms starting at 200 ms after sound onset. The laser light activated PV+ cells in AAF, which in turn suppressed the activity of other AAF cells (Figure 1n). The laser presentation during the ongoing sound did not result in any significant change in the animals’ performance (Figure 1o, p, Δ Hit rate (off-on) = 2.6±2.7%)."

(l.262-268) "Finally, we performed experiments in PV-ChR2 animals where the laser light was applied for 200 ms during the sound presentation (Figure 2n). The laser light activated PV+ cells in AAF, which in turn suppressed the activity of other AAF cells (Figure 2o). The laser presentation during the ongoing sound did not result in any significant change in the animals’ performance (Figure 2p, r, Δ Hit rate (offon) = 2.7±3.1%). Together, these experiments confirm that changing offset responses, but not sustained activity, in AAF influences behavioral performance."

5) It would be helpful to present more details about how the classifier was trained and tested (Figure 3). The neuronal classifier also does not seem to predict accurately behavior as the accuracy for the offset responses is at ~65 % (Figure 3c). It is furthermore interesting that much of behavior can be explained based on spontaneous neuronal activity, and therefore an alternative interpretation would be that cortical state (or another factor that accounts for differences in spontaneous activity between trials). This also suggests that a control as suggested in point 4 (laser presented at different time points in the stimulus) would be useful, as the classifier would predict significant reduction in performance based on suppression of activity prior to sound, and not at sound offset.

Details on how classifier was trained are now added to the methods section.

(l.827-838) "Logistic regression was implemented using the sklearn function LogisticRegression with the lbfgs solver and L2 regularization to avoid over-fitting. 8-fold cross validation was performed by leaving out a random 12,5% subset of trials to test the classifier performance, and remaining trials were used to train the classifier. A range of regularization values was tested (0.0001 to 10000 log spaced), and the one that gave the smallest error on the validation dataset was chosen as the optimal regularization parameter. The classifier accuracy was computed as the percentage of testing trials in which the animal's choice was accurately predicted by the classifier and summarized as the average across the 10 repetitions of trial subsampling. The spiking activity of each neuron was z-scored before running the logistic regression model. Trial labels were shuffled to confirm that decoding is not working for random data. This procedure was repeated 10 times. Then the average across the 10 repetitions was used to assess classifier accuracy for randomized data. "

As suggested, we also trained a classifier based on the offset responses preceded by activity suppression prior to the sound termination (Author response image 1). We found that offset responses were not significantly different between laser on and laser off trials when the activity was suppressed in the time window 200-400 ms following sound onset. As spiking at the offset was not affected, the classifier did not predict a significant reduction in the performance for the laser on trials (Author response image 1).

**Author response image 1. sa2fig1:** The classifier does not predict any reduction in performance following the suppression of AAF activity during the sound presentation. (a) Averaged PSTH (mean± SEM) of AAF neuron's response to 1 s long pure tone (9 kHz) played at 60 dB SPL during hit (blue) and miss (red) trials with laser applied for 200 ms during ongoing sound (0.2-0.4s). (b) Comparison of classifier accuracy of decoders trained and tested on offset response of AAF neurons in trials with laser-on and -off (mean± SEM), p=0.3060.

[Editors’ note: further revisions were suggested prior to acceptance, as described below.]

Essential revisions:Reviewer #2:I appreciate the authors' detailed responses to the concerns arising from the earlier review of this manuscript. There are no major concerns with the electrophysiology part of the revised manuscript.Behavior: My concerns with behavioral experiments and what the mice are actually reporting in the task remain.1) I respectfully disagree with the authors' statement that false alarm rates cannot be calculated for detection tasks. The term 'False Alarm' is derived from signal detection theory, where the subject reports that a signal was present when a signal was not actually present. In this case, the signal is the offset. If a mouse reports an offset by licking during the sound (when an offset, i.e. the signal, was not present), that outcome should be a false alarm. In the authors' task design, only correct rejections cannot be specified (to do so, one would have to define a trial duration independent of sound termination time. If the mouse withholds licking for the trial duration while the sound has not actually terminated, it would be a correct rejection). I agree with the authors that if the mouse starts licking before even the sound onset, those trials should be discarded.

We thank the reviewer for suggesting to use not only hit rate, but also other signal detection measures in quantifying mouse behavior. Following the advice of Reviewer #3, we have now calculated hit, correct rejection, miss, and false alarm rates based on trials in which the sound is presented for either 1 s or 2 s. The behavioral performance is then calculated as the ratio of desirable and undesirable outcomes, as you suggest in your next comment. We call this measure the offset detection index, or ODI:ODI=Hit+CorrectRejectionMiss+FalseAlarm Hit: percent of 1 s-long trials in which the mouse licked between 1 s and 2 s (during the reward window).

Miss: percent of 1 s-long trials in which the mouse did not lick between 1 s and 2 s (during the reward window).

False Alarm: percent of 2 s-long trials in which the mouse licked between 1 s and 2 s (during the last 1 s of sound).

Correct Rejection: percent of 2 s-long trials in which the mouse did not lick between 1 s and 2 s (during the last 1 s of sound).

Author response image 2 shows the ODI of last 6 days of training sessions and of the test sessions for PV-ChR2 animals. The increasing ODI across the sessions reflects the learning of the task.

**Author response image 2. sa2fig2:** Increasing ODI value over the training and test days.

2) "Likewise, unlike in a discrimination task, a hit rate of 50% is not a chance level in a detection task."- It is not a question of what the chance level was, but rather a question of whether the mouse was reliably detecting the offset. p(Hit) + p(FA) + p(Miss) + p(CR) = 1, and a measure of performance would take into account the ratio of desirable (p(Hit)+p(CR)) and undesirable (p(FA)+p(Miss)) outcomes. To illustrate, in Figure S2 panel h, I count 50 trials, with n(Hit) = 23, n(Miss) = 11, nFA = 16, and n(CR) = 0. In this counting, the number of desirable outcomes (23+0=23) is lesser than the number of undesirable outcomes (11+16 = 27), and would result in a d' = z(Hits) – z(FA) = 0.37. However, if trials in which animals licked during the sound (which I coded as FA) are dropped, then n(Total) = 50-16 = 34, with n(Hits) = 23 (23/34 = 67%). This is why dropping the onset lick trials and considering 30% hits as reaching criterion for training is not convincing.Other metrics to better capture the behavior might be a discrimination index as used in, for example, Schwartz and David, 2018. Here the cumulative lick probability can be used to calculate ROC curves.

As described above, an offset detection index (ODI) is now used to compare the animals’ performance for different conditions (laser on vs laser off; fast ramp vs slow ramp). Taking into account hits, misses, correct rejections, and false alarms confirms that reducing cortical offset responses with optogenetics significantly decreases the behavioral performance in sound termination detection task (Figure 1g, h; Figure 2h, i). This is in accordance with our previous conclusions made based on the analysis of the hit rates alone and strengthen our results.

In the revised manuscript, we have:

– Updated Figure 1 and Figure 2 with ODI instead of hit rates.

– Added the corresponding ODI values to the Results section (l. 128, 132, 138, 215, 229, 233, 237).

– Added the detailed description on how ODI was calculated to the methods section (l. 823-836).

– Moved the behavioral measures of hit rate previously shown in Figures 1 and 2 to Figure 1 —figure supplement 3 of the revised manuscript.

3) Page 5, lines 121 and 125: The authors use a very similar effect size change in hit rate for PV-Cre animals (4.5 +/- 2%) and for wild type animals (4.4 +/- 2.4%) to make the claim that there was a significant change in hit rate (p = 0.0264) for PV-cre animals, but no change (p=0.058) for the wild type animals (Figures 1H and 1L). The appropriate comparison in this case would be between 1H and 1L. (similar analysis must be performed comparing Figure 2I and 2M). It would be instructive if the authors could plot the Laser OFF and ON hit rates separately for the two groups (PV-Cre and wild type animals) so that one can also appreciate if there are overall differences in performance levels.

As described in the previous point, we are now using the offset detection index (ODI) to evaluate the animals’ behavior performance in the sound termination detection task. In the updated version of the manuscript, we report the effect size change of ODI for PV-ChR2 and WT animals (10 ms ramp: PV: Δ ODI=0.23±0.06, p=0.0005; WT: Δ ODI=0.15±0.11, p=0.24; 0.01 ms ramp: PV: Δ ODI=0.25±0.09, p=0.0095; WT: ODI=‑0.04±0.09, p=0.41, one sample Wilcoxon test). We also compared the general performance of the animals but did not see any significant differences in the overall performance levels (Author response image 3; 10 ms ramp, laser off: p=0.57, Kruskal-Wallis test; 0.01 ms ramp, laser off: p=0.16, Kruskal-Wallis test).

**Author response image 3. sa2fig3:** Overall performance levels in different experimental groups.

Additionally, the sizes of the error bars in Figure 1H and 1L are inconsistent with the numbers reported (why does 1H have the larger error bars despite the smaller value of 2% reported in main text)?

We thank the Reviewer for this comment. Indeed, panel H represented SD and panel L SEM, while in the main text we reported SEM values. In the updated version of the manuscript, the difference in ODI are represented as mean± SEM in Figures 1 and 2 and the corresponding values are reported in the main text.

Citing recent literature:I appreciate the authors' inclusion of two recent studies in the Discussion section. However, these studies also affect the framing of the manuscript in the introduction section.Page 2, line 39: "De-novo generation or amplification of offset responses in these areas have not been demonstrated yet." – Please rephrase and appropriately cite the Bondanelli 2021 study here.

We have now updated the introduction of the manuscript with this citation.

(l.37-44) “Offset responses in MGB and ACx are generally thought to be driven by excitatory/inhibitory inputs from IC rather than by other neural mechanisms (Kopp-Scheinpflug et al., 2018). Recently, Bondanelli et al. also argued for a role of recurrent A1 connectivity in shaping offset responses in cortex, and suggested that cortical offset responses could be generated at a higher level of recurrency (Bondanelli et al., 2021). However, de novo generation or amplification of offset responses in MGB and ACx have not been demonstrated experimentally yet (Bondanelli et al., 2019; He, 2003; Kasai et al., 2012; Yu et al., 2004).”

Page 2, lines 42 – 52: In this paragraph discussing the perceptual significance of offset responses, please appropriately discuss the Li 2021 study. Please rephrase the last sentence of this paragraph.

We have now updated the introduction of the manuscript with this citation.

(l.53-56) “Recently, it was shown that A1 transient offset responses contribute critically to encoding and perceiving sound duration (Li et al., 2021). Whether the increased neuronal activity of sound offset responses accounts for other perceptual skills is unclear.”

Page 4, line 86: Please rephrase "Changing the neuronal activity of sound offset responses without changing any other parameters of the sound response has not been tested" in light of the Li 2021 study.

It is true that sound offset responses have also been manipulated in the Li et al. 2021 study. The test used was however a duration discrimination task, and not a sound termination detection task as in our study. We have now updated the introduction of the manuscript accordingly.

(l.88-l.91) “The perceptual significance of cortical offset responses has been difficult to assess. Indeed, confounds about perceiving a sound and its termination are intricately linked. Changing the neuronal activity of sound offset responses to evaluate its contribution in sound termination detection has not been tested.”

Decoding analysis:The classifier accuracy is not significantly different between the offset and onset windows, which suggests that onset responses can be as predictive of the mouse's offset detection behavior as offset responses. Could the authors please discuss?

This is now discussed in the Results section.

(l.292-295) “These results suggest that AAF offset responses can be informative on the animal's decision emphasizing the behavioral relevance of AAF offset responses. However, the classifier accuracy was not significantly different between the offset and onset windows, suggesting that perceiving a sound and its termination are intricately linked.”

Reviewer #3:The paper represents an innovative and comprehensive body of work aimed to assess the function of cortical offset responses in the anterior auditory field in sound perception. Whereas the majority of work to date in auditory neuroscience has focused on the sound onset responses (largely due to the quick adaptation of cortical responses), the offset responses, which are present in the cortex, and especially, as the authors show, in AAF, have received less attention. The offset responses can play an important role in sound segregation and auditory scene analysis. The authors used a combination of behavioral, electrophysiological and optogenetic techniques to study the properties of cortical offset responses. The authors first test whether and how offset responses correlate and affect behavioral detection of sound offsets. They find that suppressing offset responses in AAF reduces the responses of mice to sound offsets and that there is a significant correlation between cortical responses and behavioral report. The authors next use elegant electrophysiological and manipulation methods to find that cortical offset responses have a component that is generated in the AAF and therefore not only inherited from the periphery. The authors also find that offset responses increase with sounds that have longer duration and therefore do not simply encode for silence. Behaviorally, authors provide evidence for a role of cortical offset responses in sound termination perception. Such an extensive description of these types of responses provides for a substantial advance in our understanding of cortical function.The authors have conducted additional experiments and analysis and the revised manuscript is much improved. We have only 1 outstanding concern.Responses to 1+2: We believe that it is important to consider not only the hit rate, but also other signal detection measures in interpreting mouse behavior. In fact, trials on which the mouse licked during sound may be informative, and I am not sure it's warranted to not include information about them in the analysis. One possible approach would be to compute signal detection theory measures of HR (hit rate), CR (correct rejects), M (misses rate) and FA (false alarm rate), from which one can compute d' using standard approaches.– Consider only the trials in which the sound is presented for either 1 s or 2 s.– To compute hits, compute the percent of 1s-long trials, in which the mouse licked between 1s and 2 s (during the reward window).– To compute misses, compute the percent of 1s-long trials, in which the mouse did not lick between 1s and 2 s (during the reward window).– To compute FA, compute the percent of 2s-long trials, in which the mouse licked between 1s and 2s (during the last 1s of sound).– To compute CR, compute the percent of 2s-long trials, in which the mouse did not lick between 1s and 2s (during the last 1s of sound).

We thank the reviewer for suggesting to use not only hit rate, but also other signal detection measures to quantify behavioral performance. As suggested, we have now calculated hit, correct rejection, miss, and false alarm based on trials in which the sound is presented for either 1 s or 2 s. The behavioral performance is then calculated as the ratio of desirable and undesirable outcomes, as suggested by Reviewer #2. We call this measure the offset detection index, or ODI:ODI=Hit+Correct RejectionMiss+FalseAlarmODI is now used to compare behavioral performance for different conditions (laser on vs laser off; fast ramp vs slow ramp). Taking into account hits, misses, correct rejections, and false alarms confirms that reducing cortical offset responses with optogenetics significantly decreases the behavioral performance in the sound termination detection task (Figure 1g, h; Figure 2h, i). This is in accordance with our previous conclusions made based on the analysis of the hit rates alone and strengthens our results.

In the revised manuscript, we have:

– Updated Figure 1 and Figure 2 with ODI instead of hit rates.

– Added the corresponding ODI values to the Results section (l. 128, 132, 138, 215, 229, 233, 237).

– Added the detailed description on how ODI was calculated to the methods section (l. 823-836).

– Moved the behavioral measures of hit rate previously shown in Figures 1 and 2 to Figure 1 —figure supplement 3 of the revised manuscript.

Response to 10. Please use <= 2 significant digits for p values (e.g. p = 0.5655 should be reported as p=0.57 on line 415).

We have now updated the reported p values to 2 significant digits.